# Designing Cyclic Peptides via Harmonic SDE with Atom-Bond Modeling

Xiangxin Zhou [* 1 2 3]   Mingyu Li [* 4 5]   Yi Xiao [4]   Jiahan Li [4]   Dongyu Xue [1]   Zaixiang Zheng [1]
Jianzhu Ma [4 6]   Quanquan Gu [1]

## Abstract

Cyclic peptides offer inherent advantages in pharmaceuticals. For example, cyclic peptides are more resistant to enzymatic hydrolysis compared to linear peptides and usually exhibit excellent stability and affinity. Although deep generative models have achieved great success in linear peptide design, several challenges prevent the development of computational methods for designing diverse types of cyclic peptides. These challenges include the scarcity of 3D structural data on target proteins and associated cyclic peptide ligands, the geometric constraints that cyclization imposes, and the involvement of non-canonical amino acids in cyclization. To address the above challenges, we introduce CPSDE, which consists of two key components: ATOMSDE, a generative structure prediction model based on harmonic SDE, and RESROUTER, a residue type predictor. Utilizing a routed sampling algorithm that alternates between these two models to iteratively update sequences and structures, CPSDE facilitates the generation of cyclic peptides. By employing explicit all-atom and bond modeling, CPSDE overcomes existing data limitations and is proficient in designing a wide variety of cyclic peptides. Our experimental results demonstrate that the cyclic peptides designed by our method exhibit reliable stability and affinity.

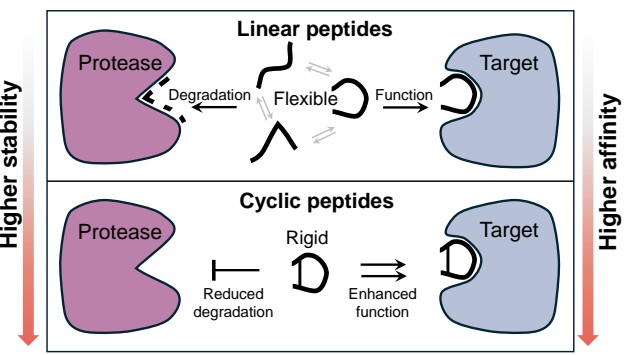

*Figure 1.* Comparative advantages of cyclic peptides over linear peptides. Linear peptides are easily degraded, whereas cyclic peptides are protected against enzyme hydrolysis, allowing them to function more effectively within the human body. Cyclic peptides generally exhibit better stability and affinity.

## 1 Introduction

Therapeutic peptides are a distinct group of pharmaceutical compounds comprising a sequence of precisely arranged amino acids (Driggers et al., 2008; Tsomaia, 2015; Zorzi et al., 2017). Peptide drugs tend to exhibit lower toxicity, enhanced biological activity, and high target specificity compared to small molecules, superior cellular permeability, lower cost, lower immunogenicity compared to antibody drugs (Driggers et al., 2008). However, conventional linear peptides suffer from a short half-life, limited stability, and susceptibility to hydrolase degradation (Tsomaia, 2015), restricting their therapeutic potential and broader use. Unlike linear peptides, as shown in Figure 1, cyclic peptides are chains of residues that typically form one or two closed loops, often incorporating non-natural residues. For example, a single loop can be created by connecting the N- and C-termini with a peptide bond or linking two internal cysteines through a disulfide bond (Camarero & Muir, 1999; Giordanetto & Kihlberg, 2014; Kale et al., 2018). Such cyclization enhances their resistance to digestive enzymes and enables them to bind protein surfaces with high affinity in more stable conformations. Traditional methods for discovering cyclic peptides involve chemical synthesis and high-throughput screening, both of which are labor-intensive and costly. This has led to adopting *in silico* approaches such as

---

[*]Equal contribution [1]ByteDance Seed (Work was done during Xiangxin's internship at ByteDance Seed.) [2]School of Artificial Intelligence, University of Chinese Academy of Sciences [3]New Laboratory of Pattern Recognition (NLPR), State Key Laboratory of Multimodal Artificial Intelligence Systems (MAIS), Institute of Automation, Chinese Academy of Sciences (CASIA) [4]Institute for AI Industry Research, Tsinghua University [5]School of Medicine, Shanghai Jiao Tong University [6]Department of Electronic Engineering, Tsinghua University. Correspondence to: Quanquan Gu <quanquan.gu@bytedance.com>.

*Proceedings of the 42$^{nd}$ International Conference on Machine Learning*, Vancouver, Canada. PMLR 267, 2025. Copyright 2025 by the author(s).

virtual screening (Zotchev et al., 2006) and de novo design (Kawamura et al., 2017; Peacock & Suga, 2021; Bhardwaj et al., 2022; Garcia Jimenez et al., 2023) to streamline the discovery process.

Cyclic peptides can adopt various structural forms depending on how their amino acid residues are linked together based on different chemical bonds and geometric distances, for example, the distance between two cysteines forming a disulfide bond typically falls from 2.0 to 2.5 Å (Fass, 2012). These structures are classified into four categories (see Figure 2) based on the atoms of residues that form the cyclic structure: (1) **Head-to-tail cyclization.** The N-terminus (head) of one amino acid forms a peptide bond with the C-terminus (tail) of another amino acid, resulting in a closed ring. (2) **Side-to-tail cyclization.** The side chain of an amino acid is linked to the C-terminus (tail) of the peptide. (3) **Head-to-side cyclization.** The N-terminus (head) of one amino acid is linked to the side chain of another amino acid. (4) **Side-to-side cyclization.** The side chain of an amino acid is linked to the side chain of another, forming a cyclic structure that does not involve the head or tail of the peptide backbone. Therefore, incorporating specific chemical and geometric constraints (such as bond lengths, angles, and atom compositions) relating to one or more sets of residues is essential for designing different kinds of cyclic peptides. Recent attempts have tried designing disulfide-linked cyclic peptides based on a post-processing strategy (Wang et al., 2024a) or head-to-tail cyclic peptides (Rettie et al., 2024) using modified position encoding in protein generative models. However, these methods only consider one specific type of cyclic peptides and do not support other types based on specific constraints. Furthermore, the availability of real-world 3D structural data for protein-ligand complexes involving cyclic peptides is limited, posing a challenge to advancing computational cyclic peptide design.

To tackle these challenges, we developed the CPSDE, which comprises two models: a harmonic-SDE-based generative structure prediction model named ATOMSDE and a residue type predictor named RESROUTER. Unlike leading works (Watson et al., 2023; Yim et al., 2023) that typically use the residue frame representation for protein design, we employ the all-atom and bond representation. Since both linear and cyclic peptides are composed of atoms and bonds, this representation allows us to model interactions at the most fundamental level. It maximizes the use of small molecule and linear peptide data while minimizing reliance on cyclic peptide data. The inclusion of bond modeling effectively addresses the geometric constraints introduced by cyclization. With our designed routed sampling method, we iteratively update both the sequence and structure by alternating between the two models, which enables the generation of all types of cyclic peptides. We highlight our main contributions as follows:

- We introduce CPSDE, the first generative algorithm, to our knowledge, capable of directly generating all types of cyclic peptides informed by the 3D structure of a protein target, paving the way for developments in peptide-based drug discovery.
- Our approach designs cyclic peptides with robust stability and affinity while maintaining high diversity, underscoring its significant potential in drug development and therapeutic innovation.
- Through case studies involving molecular dynamics simulations, we demonstrate our method's practical utility and effectiveness in drug design, reinforcing its applicability and impact in real-world scenarios.

## 2 Related Work

**All-Atom Protein Design.** Protein design traditionally involves first designing the backbone, followed by sequence design. Diffusion models (Ho et al., 2020; Song et al., 2021) have been applied to protein backbone design (Watson et al., 2023). This process is typically followed by inverse-folding models (Dauparas et al., 2022), enabling comprehensive protein design. Recently, there has been a shift towards *co-design*, where both protein sequences and structures are generated jointly (Jin et al., 2022; Luo et al., 2022; Kong et al., 2023; Lisanza et al., 2024; Campbell et al., 2024). However, all-atom[1] structure modeling is essential for comprehending protein functionality, such as protein-protein interactions. Consequently, recent research has increasingly focused on all-atom protein design to gain a more detailed and accurate understanding. To achieve full-atom antibody design, Kong et al. (2024) proposed to use a multi-channel equivariant layer to encode all-atom structures, and Martinkus et al. (2023) introduced a backbone and internal generic side chain representation. Chen et al. (2025) adopted a representation that includes amino acid type, backbone structure, and sidechain torsion angles for designing protein complexes. A notable advance in *de novo* all-atom protein design is Protpardelle (Chu et al., 2024), which suggested modeling a "superposition" over possible side-chain states and introduced the atom73 representation. This inspires us to apply all-atom structures in a similar manner for *de novo* cyclic peptide design. Unlike Protpardelle, we also incorporate bond modeling, which is crucial for ensuring successful cyclization.

**Peptide Design.** Peptides, consisting of short chains of amino acid residues, are essential in numerous biological processes due to their interactions with various target molecules, offering substantial potential in drug discovery, such as targeting undruggable proteins (Hosseinzadeh et al.,

---

[1]In some contexts, "all" in "all-atom" is interpreted to encompass all types of biomolecules, including small molecules, proteins, and nucleic acids (Krishna et al., 2024). In our work, however, we use "all-atom" with the same meaning as "full-atom", specifically to represent all heavy atoms within proteins.

2021). Traditional computational peptide design methods often rely on searching and sampling residues or motifs from chemical databases (Bhardwaj et al., 2016), which can be time-consuming and limit the diversity of designed structures (Cao et al., 2022). In contrast, deep generative models, known for their strong capability in modeling data distributions, have been applied to peptide design and have shown great potential. Several studies have explored designing peptide backbones (Boom et al., 2024), designing peptide sequences (Chen et al., 2024), or generating specific peptide structures such as $\alpha$-helices (Xie et al., 2023; 2024). Lin et al. (2025) proposed target-aware peptide sequence-structure co-design with flow matching on peptide global translation, orientation, backbone torsions, and sequences. Recent advances in full-atom protein design have significantly improved peptide generation capabilities. For example, Kong et al. (2024) employed a latent diffusion model on a latent space that encodes full-atom peptide structures. Li et al. (2025) proposed to represent peptides with backbone atoms and side-chain torsion angles, employing flow-based models to generate full-atom peptides given the protein targets. These methods face challenges in adapting to cyclic peptide design because they model protein structures at the residue level, which complicates the incorporation of covalent bonds or non-canonical amino acids essential for cyclization.

## 3 Method

In this section, we present CPSDE, a groundbreaking approach for designing cyclic peptides. It features an SDE-based generative structure prediction model, ATOMSDE, and a residue type predictor, RESROUTER, both utilizing all-atom and bond modeling. We begin by defining the cyclic peptide design task and providing an overview of SDE-based generative models in Section 3.1. In Section 3.2, we detail ATOMSDE, based on a harmonic SDE, and in Section 3.3, we describe RESROUTER, which predicts residue types based on denoised structures. Lastly, in Section 3.4, we explain how to alternate between these models through routed sampling to generate cyclic peptides.

### 3.1 Preliminaries

A peptide is a specific type of protein, generally composed of fewer than 30 amino acid residues. The type of the $i$-th residue $a_i \in \{1, 2, \ldots, 20\}$ is determined by its side-chain R group. Thus, the **all-atom** 3D structure of a peptide specifies both its sequence and structure, inspiring us to focus on generating 3D coordinates of all atoms for peptide design. Cyclic peptides, particularly their cyclization regions, often include unique inter-residue chemical bonds and occasionally non-canonical amino acids. Despite their non-canonical nature, these cyclization regions usually display distinct patterns. Additionally, providing detailed cyclization information during cyclic peptide design is essential, as it ensures

wet-lab synthesizability. Here we denote the all-atom 3D structure of the cyclic peptide as $\mathcal{P}$ and the 3D structures of the receptor (i.e., protein target) as $\mathcal{T}$. We define the chemical graph of the cyclization parts as $\mathcal{C}$, which contains atoms as nodes and chemical bonds as edges. Note that since the 3D structures of the cyclization part are unavailable, there is no information about atom positions in $\mathcal{C}$. Our final goal is to model the conditional distribution $P(\mathcal{P}|\mathcal{T}, \mathcal{C})$, i.e., generate the cyclic peptides given the 3D receptor structure and the cyclization chemical graph.

We provide basic knowledge on stochastic differential equations (SDE) and SDE-based generative models (Song & Ermon, 2019; Ho et al., 2020; Song et al., 2021). SDE-based generative models (also known as score-based generative models and diffusion models) learn the data distribution by learning to denoise. The forward SDE injects noise gradually into the data $\mathbf{x}_0 \in \mathbb{R}^d$ and constructs a diffusion process $\{\mathbf{x}_t\}_{t\in[0,1]}$, such that $\mathbf{x}_0 \sim p_0$ and $\mathbf{x}_1 \sim p_1$, where $p_0$ is the data distribution and $p_1$ is the prior distribution:

$$d\mathbf{x} = \mathbf{f}(\mathbf{x}, t)\, dt + g(t)\, d\mathbf{w}, \qquad (1)$$

where $\mathbf{f}(\cdot, t) : \mathbb{R}^d \to \mathbb{R}^d$ is a vector-valued function known as drift coefficient, $g(\cdot) : \mathbb{R} \to \mathbb{R}$ is a scalar function known as diffusion coefficient, and $\mathbf{w}$ is the standard Wiener process (also known as Brownian motion). The induced perturbation kernel $p_{0t}(\mathbf{x}_t|\mathbf{x}_0)$ is a Gaussian distribution that can be efficiently sampled. The resultant $p_1(\mathbf{x}_t|\mathbf{x}_0)$ typically approximates a Gaussian distribution $p_1(\mathbf{x}_t)$ (i.e., prior distribution), which is independent of $\mathbf{x}_0$.

The reverse SDE starts from a sample from the prior distribution, denoises the noisy sample iteratively, and finally produces a generated sample:

$$d\mathbf{x} = [\mathbf{f}(\mathbf{x}, t) - g(t)^2 \nabla_{\mathbf{x}} \log p_t(\mathbf{x})]\, dt + g(t)\, d\bar{\mathbf{w}}, \quad (2)$$

where $\bar{\mathbf{w}}$ is a standard Wiener process when time flows backwards from 1 to 0 and and $dt$ is an infinitesimal negative timestep. Typically, a neural network $\mathbf{s}_{\boldsymbol{\theta}}(\mathbf{x}_t, t)$ is used to approximate the underlying score function $\nabla_{\mathbf{x}} \log p_t(\mathbf{x})$, and it can be trained via the following score matching objective:

$$\mathcal{L} = \mathbb{E}_t[\lambda(t)\mathbb{E}_{\mathbf{x}_0}\mathbb{E}_{\mathbf{x}_t|\mathbf{x}_0}\|\mathbf{s}_{\boldsymbol{\theta}}(\mathbf{x}_t, t) - \nabla_{\mathbf{x}_t} \log p_{0t}(\mathbf{x}_t|\mathbf{x}_0)\|^2],$$

where $\lambda(t)$ is time-dependent weighting function and $t$ is uniformly sampled over $[0, 1]$.

### 3.2 ATOMSDE

To accurately model both atomic interactions and bond constraints, we initially train a docking model named ATOMSDE using protein-ligand complex data, incorporating both small molecules and peptides as ligands.

In this subsection, for brevity, given a protein-ligand complex, we assume there are $N_L$ atoms whose coordinates are $\mathbf{x}^L \in \mathbb{R}^{N_L \times 3}$ in the ligand and $N_P$ atoms whose coordinates are $\mathbf{x}^P \in \mathbb{R}^{N_P \times 3}$ in the protein $\mathcal{T}$. We denote the

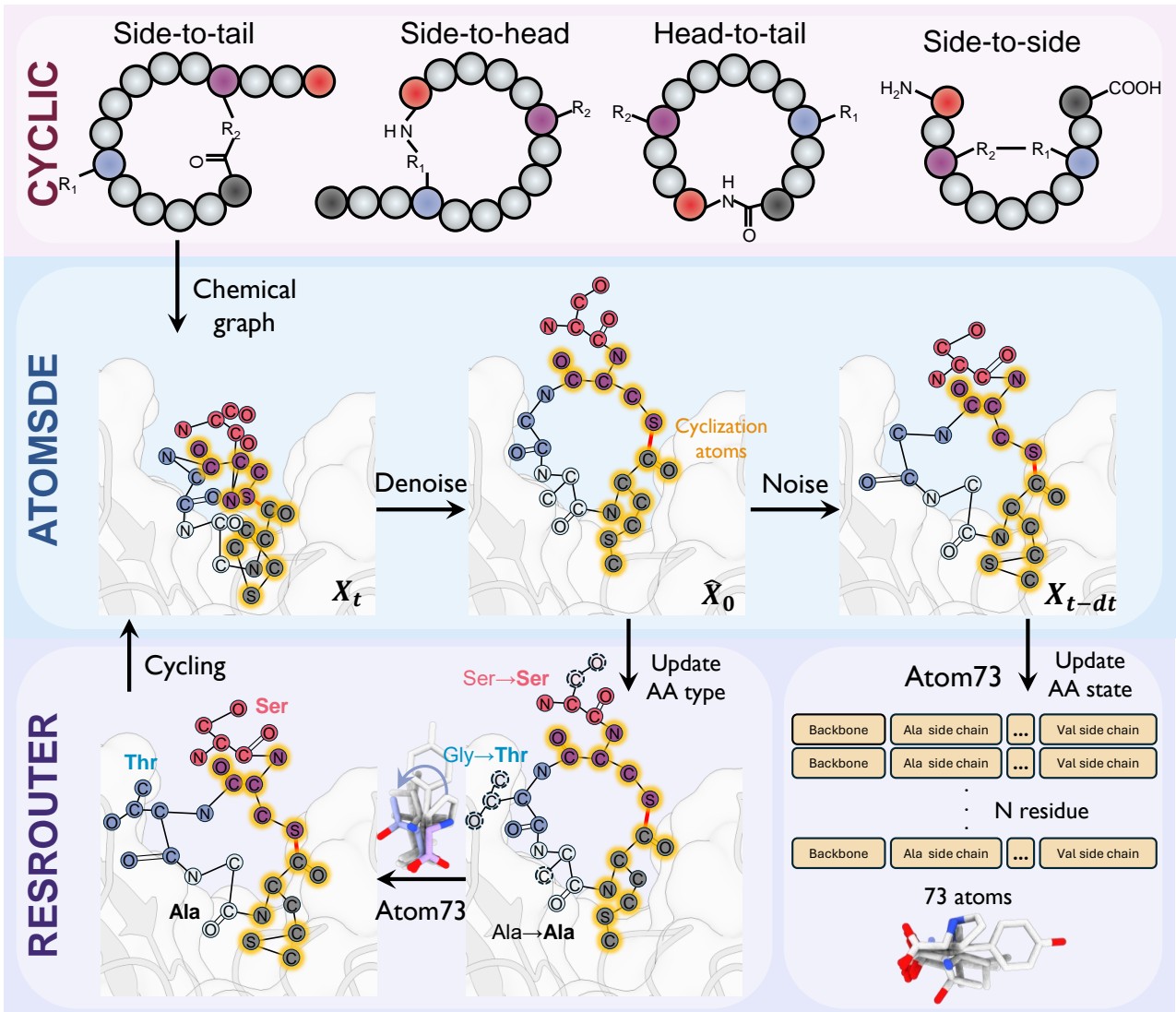

*Figure 2.* Overview of CPSDE. The generative process is structured as follows: (1) A cyclization type is initially selected, which subsequently determines the associated 2D chemical graph; (2) At time $t$, given the entire chemical graph defined by both cyclization (highlighted with a yellow shadow) and the predicted residue types, the all-atom structure is initially denoised using ATOMSDE and then re-noised in accordance with the integration step of the reverse-time SDE. The updated structures are then preserved in the Atom73 representation; (3) RESROUTER predicts the residue types not constrained by cyclization, based on the denoised structure. Consequently, the chemical graph and the all-atom structures are updated using the Atom73 representation Steps (2) and (3) are iteratively executed. The incorporation of the cyclization chemical graph, with chemical bonds as edges, ensures that the generated peptide forms a cyclic structure.

chemical graph of the ligand as $\mathcal{G}_C$, where nodes are atoms and edges are chemical bonds. We build a generative docking model ATOMSDE to learn the conditional distribution $p(\mathbf{x}^L | \mathcal{T}, \mathcal{G}_C)$.

We choose Variance Preserving (VP) SDE (Ho et al., 2020) instead of Variance Exploding (VE) SDE (Song & Ermon, 2019) for our scenario. The reason is that, when $t$ is large, VE SDE causes the noisy ligands to drift far from the receptor and introduces different spatial scales between the ligands and receptor, leading to a loss of valid interaction with the receptor. Inspired by Jing et al. (2023); Stark et al. (2023), we introduce a harmonic SDE to fully leverage the

connection information embedded in the chemical graph. We define $\mathbf{H} := \mathbf{L} + \sigma_P^{-2}\mathbf{I}$, where $\mathbf{L} = \mathbf{D} - \mathbf{A}$ is the Laplacian matrix of chemical graph $\mathcal{G}_C$, $\mathbf{D}$ is the degree matrix, $\mathbf{A}$ is the adjacent matrix, and $\sigma_P$ is a receptor-dependent scalar value. The positive definite matrix can be decomposed as $\mathbf{H} = \mathbf{P}\mathbf{\Lambda}\mathbf{P}^\intercal$ where $\mathbf{P}$ is an orthogonal matrix (i.e., $\mathbf{P}\mathbf{P}^\intercal = \mathbf{I}$) and $\mathbf{\Lambda} = \mathrm{diag}(\lambda_1, \dots, \lambda_{N_L})$ is a diagonal matrix that contains the eigenvalues. We define the $\mathcal{G}_C$-dependent forward SDE as follows:

$$\mathrm{d}\mathbf{x}^L = -\frac{1}{2}\beta(t)\tilde{\mathbf{x}}^L \, \mathrm{d}t + \sqrt{\beta(t)}\mathbf{\Lambda}^{\frac{1}{2}}\mathbf{P}^\intercal \, \mathrm{d}\mathbf{w}, \quad (3)$$

where $\beta(t)$ is a positive time-dependent scalar function that

controls the noise level along the diffusion process. The induced perturbation kernel has an analytic form as (see the derivation in Appendix F):

$$p_{0t}(\mathbf{x}^L|\mathbf{x}_0^L) = \mathcal{N}(\mathbf{x}_t^L; \mathbf{x}_0^L e^{-\frac{1}{2}\int_0^t \beta(s)\mathrm{d}s}, \mathbf{H} - \mathbf{H}e^{-\int_0^t \beta(s)\mathrm{d}s}).$$

Given a schedule that satisfies $\lim_{t\to 1}\int_0^t \beta(s)\mathrm{d}s = \infty$, the above perturbation process arrives at the prior distribution $p_1(\mathbf{x}_1^L) \propto \exp(-\frac{1}{2}\mathbf{x}_1^{L\intercal}\mathbf{H}\mathbf{x}_1^L)$ at time $t = 1$, which can be efficiently sampled. Intuitively, the anisotropic perturbation process leverages the connection information in the chemical graph $\mathcal{G}_C$. The bonded atoms are initially set close and then gradually perturbed by correlated noises.

The model is based on an SE(3)-equivariant neural network (Satorras et al., 2021; Guan et al., 2021), incorporating both a k-nearest-neighbor graph built upon the protein-ligand complex and a ligand chemical graph. This design ensures the model is aware of both protein-ligand interactions and atom connections induced by chemical bonds. We leave the details of model architecture design in Appendix G.1.

We denote the final output of the SE(3)-equivariant neural network as $\boldsymbol{D}_{\boldsymbol{\theta}}(\mathbf{x}_t^L, t)$. For simplicity, we use a simple reconstruction loss that is approximately equivariant to the score matching objective as follows:

$$\mathcal{L} = \mathbb{E}_{t, p_0(\mathbf{x}_0^L), p_{0t}(\mathbf{x}_t^L|\mathbf{x}_0^L)}[\|\boldsymbol{D}_{\boldsymbol{\theta}}(\mathbf{x}_t^L, t) - \mathbf{x}_0^L\|^2]. \quad (4)$$

The estimated score function can then be formulated as

$$\nabla_{\mathbf{x}^L} \log p_t(\mathbf{x}^L) \approx -\frac{1}{\sqrt{1 - \int_0^s \beta(s)\,\mathrm{d}s}}(\mathbf{x}^L - \boldsymbol{D}_{\boldsymbol{\theta}}(\mathbf{x}^L, t)),$$

with which we can generate the ligand poses given the ligand's chemical graph and the receptor's 3D structures by solving the reverse-time SDE as in Equation (2).

### 3.3 RESROUTER

We introduce RESROUTER that predicts the ground-truth residue type given the noisy ligands. The model architecture is similar to that of ATOMSDE. Differently, the input and output of the model are modified due to the following considerations.

With a structure prediction model in hand, we could still not be able to design cyclic peptides, since their sequence is unknown. This is a classic "chicken-and-egg" problem. This motivates us to alternately denoise the 3D structures and update the residue types. Since the residue type is determined by the side-chain R group, it will provide a shortcut for the model to predict the ground-truth residue type if the complete chemical graph of the noisy ligands is input. Thus, we remove the side chain of the canonical amino acid residues except for those that are involved in cyclization.

The model produces a hidden state (i.e., $\mathbf{h}$) for each atom. For the $i$-th residue (whose ground-truth amino acid type is $a_i$) in a peptide with $N$ residues, we aggregate the hidden

states of the backbone atoms (i.e, N-C$_\alpha$-C-O) and use a multilayer perceptron (MLP) to predict the amino acid type. The model is trained via the following objective:

$$\mathcal{L} = \sum_i^N -\log p_\phi(a_i|\boldsymbol{D}_{\boldsymbol{\theta}}(\mathbf{x}_t^L, t), \mathcal{G}_C, \mathcal{T}, t), \quad (5)$$

where $p_\phi$ is the model-induced probability for the residue type. ATOMSDE (i.e, $\boldsymbol{D}_{\boldsymbol{\theta}}$) is first pretrained and then fixed during the training of RESROUTER.

### 3.4 Routed Sampling for Cyclic Peptide Design

With trained ATOMSDE and RESROUTER, we can iteratively update both the sequence and structure through alternate calls to the two models, enabling the generation of all types of cyclic peptides, as shown in Figure 2.

Given the number of residues within the peptide and the cyclization information, the atoms in a cyclic peptide can be categorized into two classes: The first class is known in terms of their chemical graph (though their 3D structures are not determined), comprising all backbone atoms and the atoms constrained by cyclization. The second class is unknown, consisting of the side chains of canonical amino acid residues not constrained by cyclization. Without loss of generality, consider a specific type of side-to-tail cyclization where the sulfur atom (S) in the side chain of the $i$-th residue and the carboxylic acid (COOH) group of the C-terminus engage in chemical reactions to form a C-S thioester bond. In this scenario, the chemical graphs induced by all backbone atoms (since all canonical amino acid residues share the pattern N-C$_\alpha$-C-O) and all atoms of the cysteine (CYS) providing the sulfur for the C-S bond that forms the cyclic structure are known. Conversely, the chemical graph between other atoms, specifically the side chains of residues except for the aforementioned CYS residue, remains unknown. In the remainder of this paper, we refer to the atoms associated with the known chemical graph due to cyclization as *cyclization-constrained* atoms, and the others as *free-residue* atoms.

Inspired by Chu et al. (2024), for the free-residue part, we maintain an atom73 state (i.e., coordinates), a "superposition" for each residue, where all possible amino acid types for this residue share the backbone atoms and C$_\beta$, while possessing unique side chain atoms. For the cyclization-constrained part, we maintain a cyclization-constrained state separately. The general idea of routed sampling is to switch to different collapsed (i.e., specific) atom states for the free-residue part and assemble a new valid chemical graph based on the predicted residue type and cyclization information at each step of solving the reverse-time SDE. Specifically, at each step, ATOMSDE denoises the atom coordinates,

In practice, for both cyclization-constrained atoms and free-residue atoms, we maintain both denoised states and current states. This approach is needed because, during the sampling process, the cyclization-constrained atoms and backbone atoms are constantly present, while the side-chain

atoms of the free residues are sometimes sampled, resulting in them potentially not being adequately updated. Thus, we reuse the previous denoised structure to solve the SDE and align the time (or noise level) of all sampled atoms to the same point. This alignment is necessary because the ATOMSDE expects all input atoms to be at the same noise level, especially when a residue type is sampled after not being sampled for a few steps. Please refer to Appendix E for more details.

Notably, the two models, ATOMSDE and RESROUTER, do not account for the atom partition introduced by amino acid residues, as they model atoms and bonds at the most fundamental level. The concept of amino acid residues is introduced only during routed sampling to ensure that the free-residue part remains a canonical residue rather than an arbitrary molecule.

## 4 Experiments

### 4.1 Experimental Setup

**Dataset.** We have curated two datasets of protein-ligand complexes featuring small molecules and peptides as ligands, respectively. All complexes with atoms whose elements are beyond {C, N, O, F, S, Cl, Se, Br} are not included. The small molecule dataset is sourced from PDB-Bind (Wang et al., 2005) and has 14,348 protein-ligand complexes. The peptide dataset is derived from RCSB PDB (Burley et al., 2023), Propedia (Martins et al., 2023) and PepBDB (Wen et al., 2019). It comprises 20,033 protein-ligand complexes, featuring peptide ligands composed of fewer than 30 residues. Samples are clustered by receptor sequence identity of 0.3 to split the dataset into training and validation sets. To train ATOMSDE, we utilize the curated small molecule dataset and a subset of the peptide dataset containing ligands with fewer than 200 heavy atoms. To train RESROUTER, we use the whole peptide dataset. Please refer to Appendix D for more details about data.

**Baselines.** As our method is the first cyclic peptide design method based on generative models, we compare our approach with various established methods for linear peptide design: **RFDiffusion** (Watson et al., 2023) generates protein backbones, and sequences are later predicted by ProteinMPNN (Dauparas et al., 2022); **ProteinGenerator** (Lisanza et al., 2024) improves RFDiffusion by jointly sampling backbones and corresponding sequences; **PepFlow** (Li et al., 2025) is a flow-based full-atom peptide generative model that generates the translation, rotation, and side-chain torsion angles of each residue frame within a peptide; **PepGLAD** (Kong et al., 2024) is a full-atom peptide design method that utilizes a latent diffusion model.

**Evaluation.** We use Rosetta (Chaudhury et al., 2010) to compute the total energy of reference ligands, linear peptides designed by baseline methods, and cyclic peptides engineered by our approaches. This energy measurement

serves as an indicator of the stability (or rationality) of a peptide's 3D binding pose. Hence, we define this energy metric as **Stability**. We also evaluate the interface binding energy, a crucial metric that indicates the binding affinity of the ligand peptide to its receptor. This assessment is essential for evaluating the ligand peptide's functionality, particularly when designing peptides for therapeutic applications. We denote this type of energy as **Affinity**. We also report **Diversity**, the average of one minus the pair-wise TM-Score (Zhang & Skolnick, 2005) among the designed peptides, reflecting structural dissimilarities. As the reference ligands for the targets in the test set are linear, we do not report metrics that necessitate a reference sequence or structure, such as Amino Acid Recovery (AAR) and Root Mean Square Deviation (RMSD). We selected 100 protein pockets with a large volume for testing. Specifically, these targets have receptors with more than 1,000 surrounding atoms around the reference peptide ligands, serving as reliable indicators of adequate volume. For each target, all peptide design methods are used to generate a batch of peptides, from which we select the most promising (i.e., lowest energy) peptide ligand. We design four types of cyclic types (head-to-tail, head-to-side, side-to-side, and side-to-tail) using our methods, respectively, and we also report the mixed results, where the best promising peptide ligands might exhibit different cyclization types. We then report the average and median metrics across all targets. Please refer to Appendix H for more details. This evaluation strategy mirrors practical drug design scenarios where the leading ligand candidates are identified for advancement to the subsequent stages of drug development.

### 4.2 Main Results

The results are shown in Table 1. Among all co-design methods, our method exhibits superior energy performance in terms of both stability and affinity and also best diversity. Interestingly, we find that head-to-tail and head-to-side cyclic peptides show better performance than side-to-tail and side-to-side cyclic peptides. This might be due to the training dataset where C-N bonds (main covalent bonds that form head-to-tail and head-to-side cyclic structures) are more frequent than S-S and C-S bonds (main covalent bonds that form side-to-tail and side-to-side cyclic structures). This aligns with the fact that C-N bonds are generally more stable than S-S and C-S bonds in the physical world. Among all methods, RFDiffusion shows the best energy performance but low diversity, as it tends to generate $\alpha$-helices in a certain pattern. See Appendix H.5 for ablation studies.

### 4.3 Case Studies

Here, we demonstrate how CPSDE can be seamlessly integrated into real-world cyclic peptide design pipelines and discuss two scenarios: design of SMYD2 peptide inhibitors via head-to-tail cyclization and SET8 inhibitors via side-to-side cyclization.

*Table 1.* Summary of properties of reference peptides, linear peptides designed by baseline methods, and cyclic peptides designed by CPSDE. (↓) / (↑) denotes a smaller / larger number is better.

| Method | Co-Design | Peptide Type | Stability (↓) | | Affinity (↓) | | Diversity (↑) |
| --- | --- | --- | --- | --- | --- | --- | --- |
| | | | Avg. | Med. | Avg. | Med. | |
| Reference | N/A | Linear | -672.53 | -634.71 | -85.03 | -78.70 | N/A |
| RFDiffusion | ✗ | Linear | -633.51 | -607.82 | -70.30 | -61.35 | 0.55 |
| ProteinGenerator | ✓ | Linear | -576.39 | -554.70 | -46.98 | -40.39 | 0.58 |
| PepFlow | ✓ | Linear | -576.16 | -498.31 | -47.88 | -42.40 | 0.70 |
| PepGLAD | ✓ | Linear | -359.44 | -310.33 | -45.06 | -38.56 | 0.79 |
| CPSDE | ✓ | Head-to-tail Cyclic | -568.04 | -519.66 | -50.86 | -46.62 | 0.79 |
| CPSDE | ✓ | Head-to-side Cyclic | -564.81 | -508.88 | -49.81 | -44.16 | 0.79 |
| CPSDE | ✓ | Side-to-tail Cyclic | -547.14 | -502.20 | -46.92 | -37.25 | 0.78 |
| CPSDE | ✓ | Side-to-side Cyclic | -537.09 | -479.94 | -49.73 | -44.71 | 0.79 |
| CPSDE | ✓ | Mix | -580.67 | -527.80 | -55.71 | -48.42 | 0.79 |

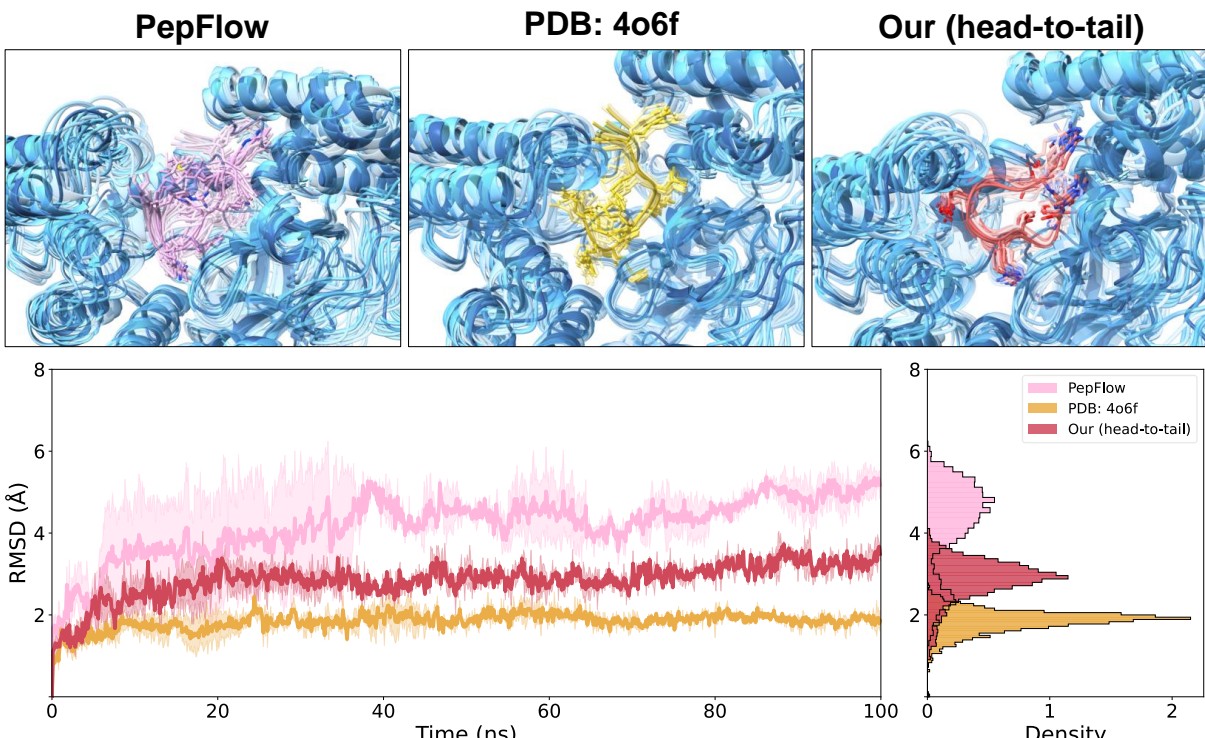

*Figure 3.* Discovery of new SMYD2 cyclic peptide inhibitors via applications of CPSDE. Upper: Visualization of conformational ensembles for the designed peptides sampled by MD. Bottom: RMSD analysis of all heavy atoms within the designed peptides.

**Design of SMYD2 peptide inhibitors via head-to-tail cyclization.** SMYD2 is an oncogene that critically regulates tumor-related signaling pathways, making it an attractive target for cancer therapy (Zheng et al., 2022). In particular, SMYD2 attenuates estrogen signaling by weakening ERα-dependent transactivation (Zhang et al., 2013). Insight into this interaction comes from the SMYD2–ERα co-crystal structure (PDB: 4O6F), which reveals ERα in a U-shaped conformation nestled within SMYD2's deep binding pocket (Jiang et al., 2014). The distance between ERα's N-terminal (head) and C-terminal (tail) in this complex is only 5.9 Å, a finding that has inspired the design of rigid head-to-tail cyclic peptides to capitalize on this proximity and enhance binding affinity.

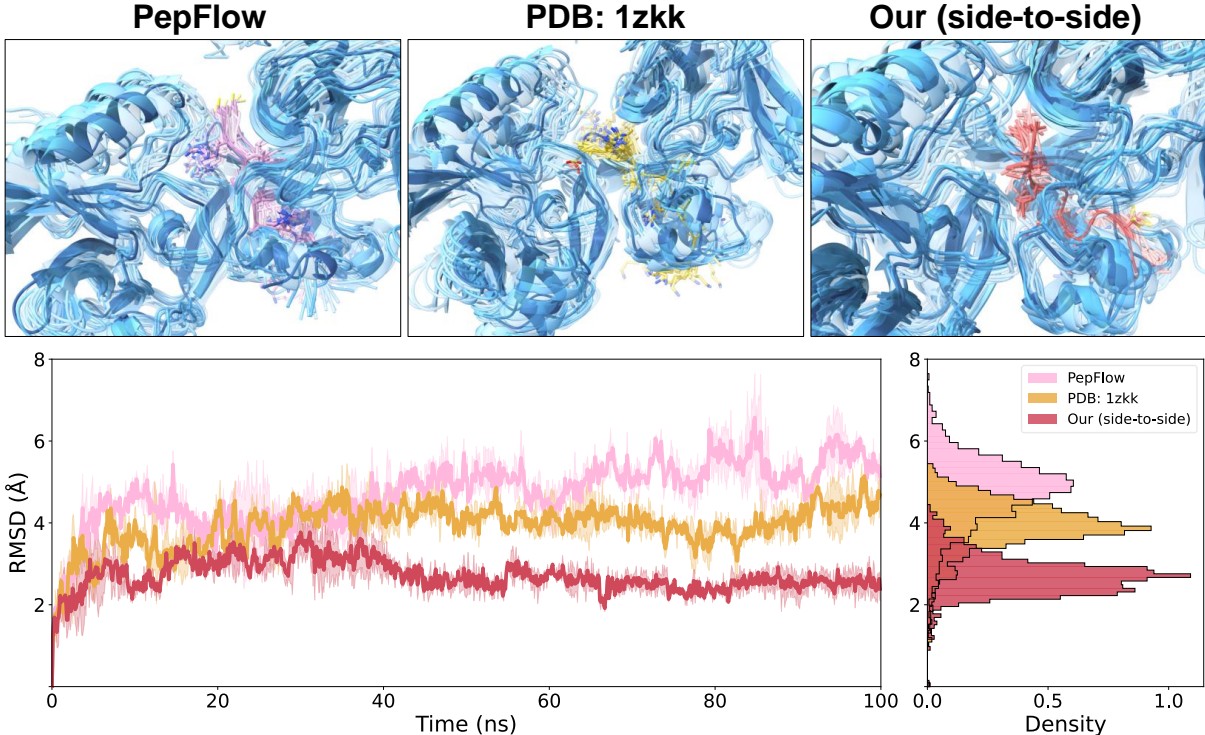

*Figure 4.* Discovery of new SET8 cyclic peptide inhibitors via applications of CPSDE. Upper: Visualization of conformational ensembles for the designed peptides sampled by MD. Bottom: RMSD analysis of all heavy atoms within the designed peptides.

Herein, CPSDE was employed to design new SMYD2 inhibitors by generating head-to-tail cyclic peptides. The binding site of ERα (PDB: 4o6f) was selected as the active pocket, and 8 cyclic peptides were generated through head-to-tail cyclization. All peptides demonstrated favorable Rosetta affinity scores (Figure 10), with H2T-6 achieving the most favorable score of -33.9 kcal/mol. To further accurately assess its binding stability and affinity, 100 ns molecular dynamics (MD) simulations were performed on the H2T-6-SMYD2 complex. More details on the system preparation and simulation protocol of MD are in Appendix H.6. The simulations also included the crystallized linear peptide (ground truth) and the PepFlow-generated linear peptide for comparison. Each simulation was repeated twice to ensure consistency. As shown in Figure 3, the linear peptide generated by PepFlow exhibited higher flexibility, with an average peptide RMSD of 4.59 Å over the last 50 ns of equilibrated trajectories. In contrast, H2T-6 displayed an average peptide RMSD of 3.05 Å, comparable to the ground truth linear peptide (1.92 Å). More importantly, binding free energy analysis using MM-PBSA (Wang et al., 2019) revealed that H2T-6 had the highest binding affinity (-24.02 kcal/mol), outperforming both the ground truth peptide (-19.00 kcal/mol) and the PepFlow-generated peptide (-7.26 kcal/mol). These results suggest that H2T-6 maintains a stable binding conformation which also exhibits high binding affinity, indicating its potential as a candidate SMYD2

inhibitor for further investigation.

**Design of SET8 peptide inhibitors via side-to-side cyclization.** SET8 is the only lysine methyltransferase that specifically catalyzes the methylation of histone H4 at the 20th lysine (Qian & Zhou, 2006). SET8-mediated protein modifications are involved in numerous physiological processes, and its dysregulation is closely linked to various human diseases, particularly cancer development and prognosis (Yang et al., 2021).

Again, we applied CPSDE to design seed cyclic peptide inhibitors targeting SET8. The 3D structure of SET8 in complex with an H4 peptide (PDB: 1zkk) was used, with the H4 binding site selected as the active pocket (Couture et al., 2005). To explore an alternative cyclization approach, we employed a widely used side-to-side strategy to generate 8 cyclic peptides. Likely, we picked up the top cyclic peptide, S2S-4 (see Figure 11), based on its Rosetta affinity score, and performed 100 ns MD simulations with 2 repeats, along with two reference linear peptides. The RMSD and MM-PBSA analyses revealed that S2S-4 not only demonstrated a significantly lower peptide RMSD (2.54 Å) compared to the reference linear peptides (ground truth: 4.06 Å; PepFlow: 5.23 Å), but also exhibited a lower binding free energy (S2S-4: -12.48 kcal/mol; ground truth: -6.39 kcal/mol; PepFlow: -9.26 kcal/mol). These results indicate that S2S-4 might serve as a potential candidate for later studies on SET8 inhibition.

# 5 Conclusion

In conclusion, CPSDE is a generative algorithm capable of producing diverse types of cyclic peptides given 3D receptor structures, thereby paving the way for advancements in peptide-based drug discovery. Our approach enhances drug development by designing stable and high-affinity cyclic peptides. Case studies supported by molecular dynamics simulations validate its practical utility and real-world applicability. Limitations and future work are discussed in Appendix I.

## Acknowledgments

We thank anonymous reviewers for their insightful feedback. We would like to extend our gratitude to Yi Zhou and Lihao Wang for their valuable feedback on our methodology, as well as to Ellen Wang, Tianze Zheng and Wen Yan for their assistance with the molecular dynamics simulations.

## Impact Statement

CPSDE contributes to the advancement of computational biology, particularly in the field of cyclic peptide design, by addressing key challenges that have hindered progress in this area. By integrating generative structure prediction and sequence prediction, our approach enables the design of stable and high-affinity cyclic peptides, offering valuable applications in drug discovery and therapeutic development. While our primary focus is on the positive applications of CPSDE, such as developing novel peptide-based treatments, we acknowledge the ethical considerations associated with any powerful generative tool. There is a potential risk of misuse, including the design of harmful bioactive compounds. To mitigate such concerns, we emphasize the responsible use of our method and encourage the scientific community to apply it for constructive and beneficial purposes. Our commitment to ethical research practices ensures that CPSDE serves as a force for innovation in pharmaceutical development while upholding societal safety and integrity.

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

## A    Introduction of Cyclic Peptides

Peptides have shown promising capability as therapeutics for protein targets where small molecules struggle to bind, due to their ability to modulate protein-protein interactions (Hosseinzadeh et al., 2021; Zorzi et al., 2017). However, traditional linear peptides are usually polar due to exposed acids and amines in terminals, limiting their membrane permeability and proteolytic stability, and also restricts their administration options in drug development (Buckton et al., 2021; Merz et al., 2024).

The peptide cyclization is capable of enhancing the conformational stability, increasing binding affinity and specificity for targets (Zorzi et al., 2017). This has fueled the growing interest in cyclic peptide research as efficient therapeutics. Figure 5 shows two cyclic peptide drugs approved in recent 20 years. According to Sharma et al. (2023); Fang et al. (2024); Costa et al. (2023), cyclic peptides can be classified by their cyclization strategies: head-to-tail cyclization (between N- and C-termini); side-to-side cyclization (between two side chains, including peptide stapling, which stabilizes $\alpha$-helical structures) (Vinogradov et al., 2019); head-to-side and side-to-tail cyclization (between a terminal and side chain); and polycyclization (with multiple cycles).

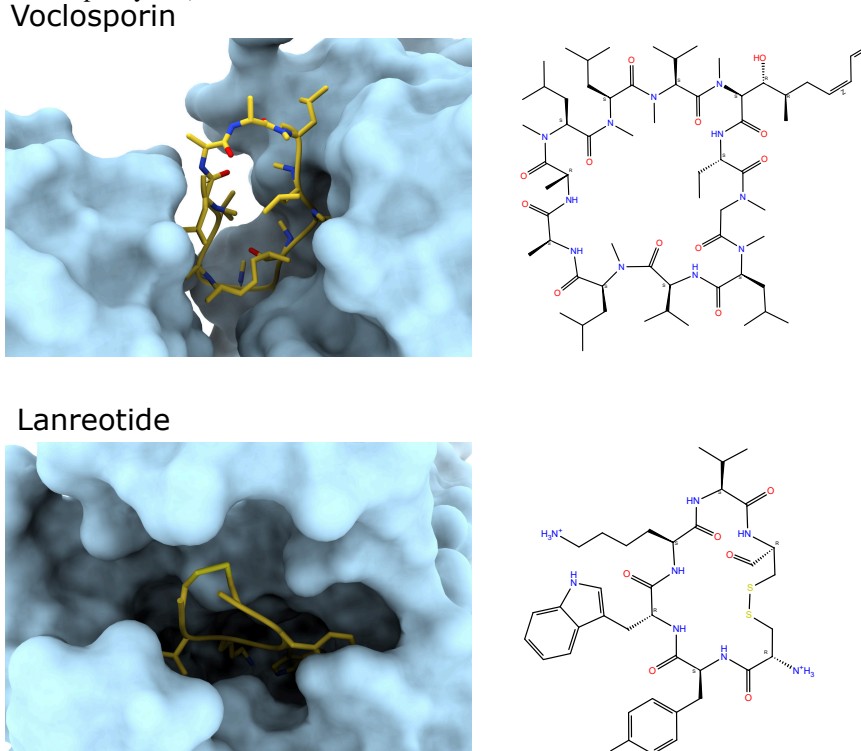

*Figure 5.* Two examples of cyclic peptide drugs. Voclosporin (van Gelder et al., 2022), an analog of ciclosporin, is an immunosuppressant used to treat lupus nephritis. Lanreotide (Caplin et al., 2014), an analog of somatostatin, is an oncology drug that inhibits growth hormone release and is used to manage carcinoid syndrome.

## B    Extend Related Work

**Protein-ligand Docking.** Protein-ligand docking aims to predict the conformation of protein-ligand complexes, playing a crucial role in drug discovery and design. Compared to traditional score-based or template-based docking methods (Eberhardt et al., 2021; Friesner et al., 2004), deep learning-based approaches are faster while achieving comparable accuracy. A common strategy involves leveraging equivariant or invariant graph neural networks to model both protein and ligand and predict ligand atom positions (Lu et al., 2022; Stärk et al., 2022; Pei et al., 2024; Qiao et al., 2024). Other approaches, such as Uni-Mol (Zhou et al., 2023) and RoseTTAFold All-Atom (Krishna et al., 2024), use positional encoding and attention layers for atom-level representations and predictions. Unlike structure prediction models, Diffdock (Corso et al., 2023) introduces the generative docking framework that views docking as a conditional generation problem, and uses a diffusion model for end-to-end ligand generation. SurfDock (Cao et al., 2024) further incorporates protein surface graphs to refine docking predictions. More recent studies have emphasized flexible docking models, which consider protein conformational variability in docking process. DynamicBind (Lu et al., 2024) and ReDock (Huang et al., 2024) extend the

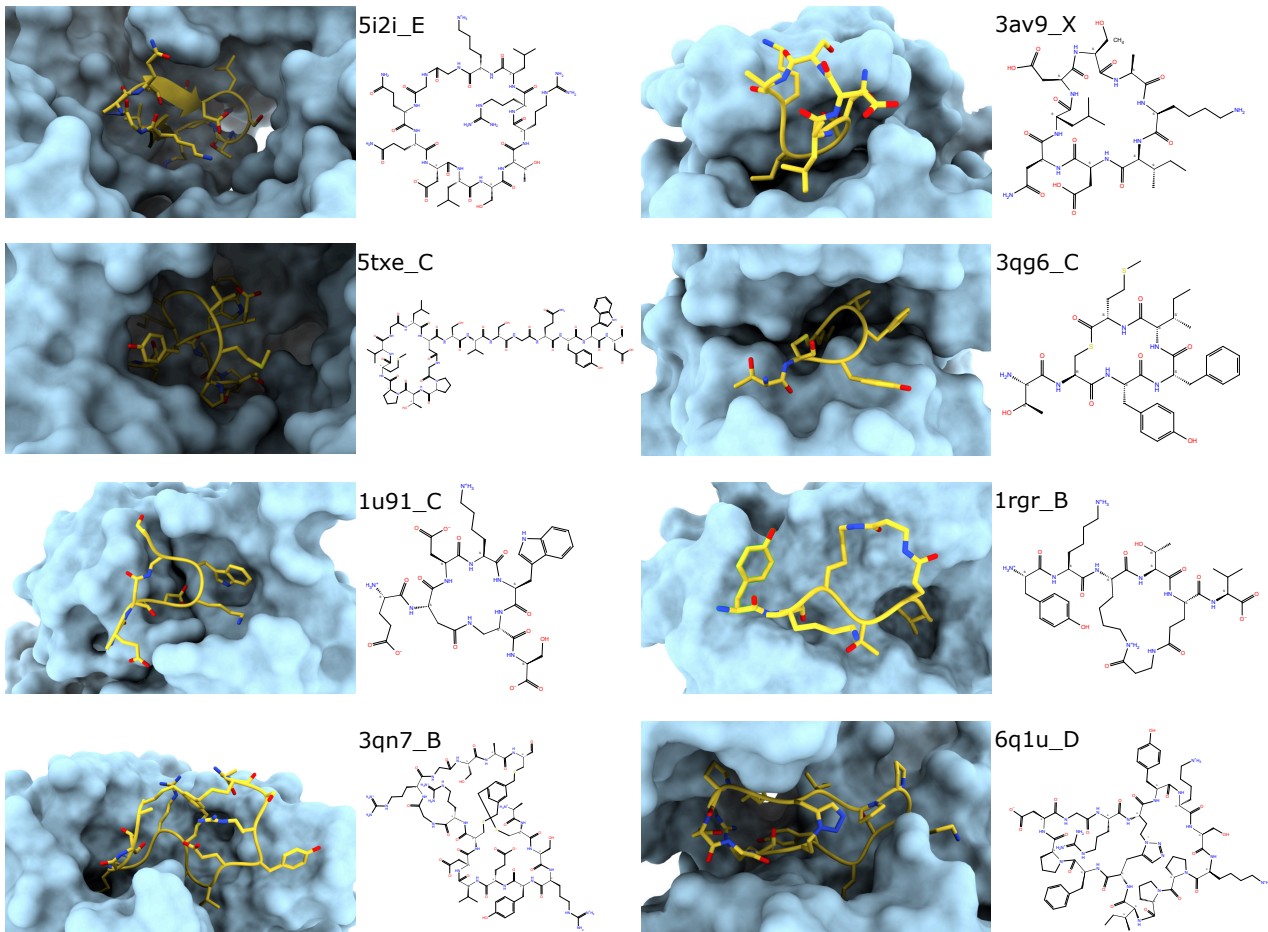

*Figure 6.* Examples of 3D structures of cyclic peptides and their chemical graphs. The cyclization structures are highlighted in pink. The corresponding cyclization types are: 5I2I_E: head-to-tail; 3AV9_X: head-to-tail; 5TXE_C: head-to-side; 3QG6_C: side-to-tail; 1U91_C: side-to-side; 1RGR_B: side-to-side (stapling); 3QN7_B and 6Q1U_D: polycyclization.

generative docking paradigm by incorporating diffusion-based modeling of protein structures, enabling joint optimization of both protein and ligand poses. Guan et al. (2025) proposed a novel molecular docking framework that simultaneously considers multiple ligands docking to a protein, enhacing molecular docking accuracy.

**Protein Sequence Design.** Designing protein sequences that fold into desired structures, known as inverse folding, is a fundamental task in protein engineering. Deep learning methods can effectively predict protein sequences given structural priors. These models generally follow a two-stage paradigm comprising a structure encoding module and a sequence prediction module. Several models, including GVP (Jing et al., 2021), ESM-IF (Hsu et al., 2022), ProteinMPNN (Dauparas et al., 2022), and Ingraham et al. (2019), utilize graph neural networks (GNNs) for structural encoding and employ autoregressive decoding for sequence prediction. PiFold (Gao et al., 2023b) adopts a similar encoding strategy but directly classifies amino acids for each node. GRADE-IF (Yi et al., 2024) formulates the inverse folding problem as learning the conditional distribution of protein sequence given the backbone structure, and employs a discrete diffusion model on protein graph to predict the amino acid type on each node. Besides graph-based approaches, Protein Language Models (PLMs) (Lin et al., 2023; Rives et al., 2021) offer an alternative strategy for inverse folding. Zheng et al. (2023) integrates PLMs as sequence decoders for encoded protein graphs, while Gao et al. (2023a) and Mao et al. (2024) incorporate ESM-2 (Lin et al., 2023) embeddings to refine sequence predictions. ProGen (Madani et al., 2023) shows an autoregressive language model trained on massive protein sequence data can utilize property tags for controllable sequence generation. Wang et al. (2024b) demonstrate that discrete diffusion serves as a more principled probabilistic framework for large-scale protein language modeling. The resulting diffusion protein language model (DPLM) excels in both sequence generation and representation learning. DPLM-2 (Wang et al., 2025) further extends this discrete diffusion-based paradigm by incorporating tokenized 3D structures thereafter, enabling structure-sequence co-generation as well as any-to-any conditional generation

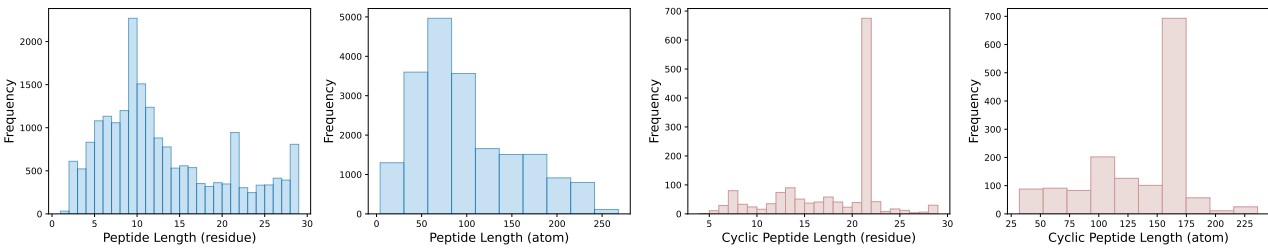

*Figure 7.* Statistics on peptide and cyclic peptide lengths in our dataset.

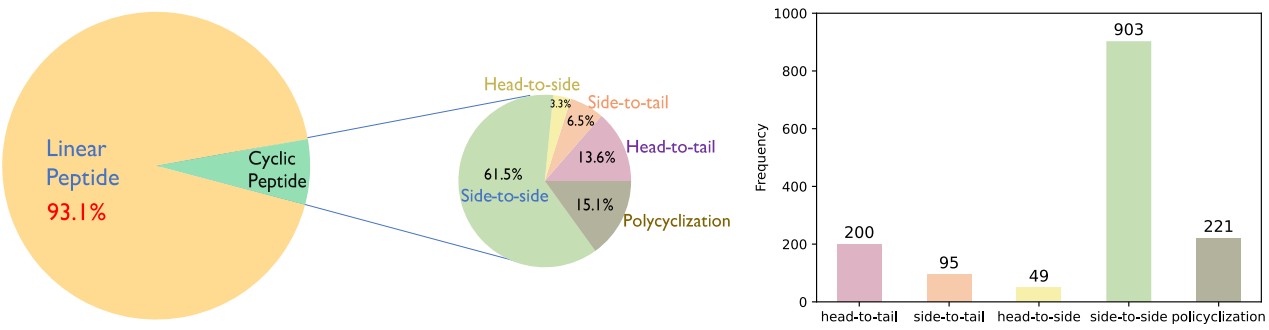

*Figure 8.* Analysis of cyclic peptide distribution and the proportions of the five cyclization types.

with multimodal generative protein language models.

**Cyclic Peptide Design.** Although computational cyclic peptide design is an innovative research area, there are already recent studies on cyclic peptide design that are significantly different from our approach (Tang et al., 2025; Rettie et al., 2024). The approach proposed by Tang et al. (2025) is a ligand-based drug design (LBDD) method that models the sequence of cyclic peptides using discrete diffusion, optimized by multiple reward functions. It does not explicitly incorporate the 3D structure of target proteins, whereas our structure-based drug design (SBDD) method directly designs ligands based on 3D target structures. Rettie et al. (2024) uses modified RoseTTAFold (Baek et al., 2021) and RFdiffusion (Watson et al., 2023) with cyclic relative positional encoding to generate macrocyclic backbones.

## C Protein all-atom representation

**Atom14 representation.** The atom14 encoding efficiently represents residues using 14 columns to capture atom content, with 14 being the maximum number of heavy atoms in the 20 canonical amino acids (e.g., Tryptophan). Empty strings are padded in atom14 representation when a residue contains fewer than 14 atoms.

**Atom37 representation.** The atom37 encoding is a fixed-dimension representation where each of the 37 heavy atom types in canonical amino acids is assigned a unique position in an array. For amino acids missing certain atoms, padding is used to maintain a consistent length. The atom37 format includes the following atoms: ['N', 'CA', 'C', 'CB', 'O', 'CG', 'CG1', 'CG2', 'OG', 'OG1', 'SG', 'CD', 'CD1', 'CD2', 'ND1', 'ND2', 'OD1', 'OD2', 'SD', 'CE', 'CE1', 'CE2', 'CE3', 'NE', 'NE1', 'NE2', 'OE1', 'OE2', 'CH2', 'NH1', 'NH2', 'OH', 'CZ', 'CZ2', 'CZ3', 'NZ', 'OXT'].

**Atom73 representation.** The atom73 representation, introduced by Chu et al. (2024), indexes the 'N', 'CA', 'C', 'CB', and 'O' atoms, then independently encodes each amino acid's side chains. Backbone atoms share the same position, while side-chain atoms are assigned to specific residue types. For example, the 'NE2' atom in the 'GLN' residue is recorded as 'Q-NE2' (where Q is the one-letter code for 'GLN') and occupies a unique column in the atom73 encoding.

## D Peptide Dataset

**CyclicPepedia.** The CyclicPepedia dataset (Liu et al., 2024) contains 9,744 cyclic peptides from multiple sources. It serves as a knowledge database with information on categorization, structural characteristics, pharmacokinetics, physicochemical properties, patented drug applications, and key publications. However, most peptide conformations are generated by RDKit, with only 1,325 cyclic peptides having original 3D structures and 61 containing complex structures. The remaining data only include fingerprints, limiting its application in structure-based drug design.

**PPBench2024.** PPBench2024 (Lin et al., 2025) is a protein-peptide binding dataset sourced from Burley et al. (2023), Martins et al. (2023), and Weng et al. (2020). It selects complexes with more than two chains and an interaction distance

under 5 Å. Complexes with peptides shorter than 30 amino acids are then filtered. Non-peptide molecules and peptides with unusual bond lengths or non-amino acid functional groups are also excluded. This results in a total of 15,593 protein-peptide pairs.

**PepBench.** PepBench is a curated benchmarking dataset designed to train and evaluate protein-peptide binding models in PepGLAD (Kong et al., 2024). The training data is sourced from (Berman et al., 2000), while the test set is adopted from Tsaban et al. (2022). To ensure quality, peptides are restricted to lengths between 4 and 25 residues, and receptors to more than 30 residues. Complexes with over 90% sequence similarity are excluded to reduce redundancy. Furthermore, clustering at 40% sequence identity is applied to separate training and test data, removing any training complexes that share clusters with the test set. The final dataset comprises 4,157 training complexes, 114 for validation, and 93 for testing.

**PepFlow.** The dataset introduced by Li et al. (2025) combines data from Wen et al. (2019) and Wei et al. (2024), resulting in 8,365 protein-peptide complex structures. Peptides are filtered to include those with lengths ranging from 3 to 25 amino acids and are clustered at 40% sequence identity. From these clusters, 158 structures are selected for the test set, ensuring each cluster has 10 to 50 members. The remaining data is used for training and validation.

**Our curation.** Our curation has been roughly introduced in Section 4.1. A unique requirement of our method is converting Structured Data Files (SDF files) to PDB files, as the chemical bond information in PDB format is often incomplete. We found that RDKit [2] does not always accurately produce chemical bonds during conversion, as it determines bond existence and type based on pairwise atom types and distances. Therefore, we leverage the Chemical Component Dictionary (CCD) [3], which describes all residue and small molecule components found in PDB entries, to collect all intra-residue bond information. We use default peptide bonds to connect canonical residues with continuous residue indices and the bonds stored in the CONECT information to recover bonds between non-canonical residues and other residues.

## E   Routed Sampling

The general idea of the proposed routed sampling is that the sequence and structure are alternately updated in the generative process. We present its details in Algorithm 1. In this subsection, we introduce some notations without rigorous definitions but maintain clarity, as we provide detailed comments after the lines in the algorithm. The sampling algorithm is inspired by Chu et al. (2024). However, we innovatively introduce a dynamic chemical graph to make the all-atom peptide design compatible with cyclic structures, especially non-canonical covalent bonds and residues. Several critical functions used in Algorithm 1 will be discussed as follows:

- "Extract" and "Cache": As introduced in Section 3.4, due to cyclization, we can categorize atoms into two types: constrained atoms and free-residue atoms. We maintain the atom73 states and individual atom states for these types, respectively. Two functions are employed to extract and store atom coordinates based on masks determined by residue types or constrained atom indices.
- "Assemble": Each time the residue types are updated, the corresponding side-chain chemical graphs are likewise updated. Consequently, we reassemble each residue's chemical graph along with the cyclization chemical graph into a complete peptide chemical graph to serve as input for the models.
- "Supgraph": As introduced in Section 3.3, to avoid residue type information leakage from the side-chain chemical graphs, we remove the side-chain atoms within the free residues from the current peptide chemical graph.
- "UpdateTime" and "AlignTime": As mentioned in Section 3.4, due to the sampling mechanism, updates of side-chain atoms within free residues might not be continuous. In other words, the coordinates of these atoms are occasionally updated, resulting in atoms having different times (or noise levels). Therefore, "UpdateTime" is introduced to store the individual time, and "AlignTime" is introduced to align the atom coordinates from different times to the same point. It's important to note that in "AlignTime", no model is involved as the previously cached denoised structures are reused.

## F   Proof of Prior Distribution Induced by Harmonic SDE

The difference between the widely-used SDE in SDE generative models and our introduced harmonic SDE is that the our perturbation process is anisotropic. Hence, here we provide the derivation of the prior distribution induced by the harmonic SDE in a similar proof by Song et al. (2021).

In Equation (3), we define the $\mathcal{G}_C$-dependent forward SDE as follows:

$$d\mathbf{x}^L = -\frac{1}{2}\beta(t)\mathbf{x}^L \, dt + \sqrt{\beta(t)}\mathbf{\Lambda}^{\frac{1}{2}}\mathbf{P}^{\mathsf{T}} \, d\mathbf{w},$$

---

[2] https://www.rdkit.org/
[3] https://www.wwpdb.org/data/ccd

---

**Algorithm 1** Routed Sampling

---

**Input:** number of residues within the ligand peptide $N$, cyclization Type $O$, 3D receptor structures $\mathcal{T}$, SDE solver time interval $dt$, infinitesimal constant $\epsilon$

**Output:** cyclic peptide with its all-atom coordinates $\mathbf{x}_0$, chemical graph $\mathcal{G}_0$, amino acid sequence $\mathcal{A}_0$

1: $[\mathbf{X}_1, \mathbf{x}_1^O] \leftarrow \text{HarmonicPrior}(N, O, \mathcal{T})$     $\triangleright$ Initialize time-dependent atom73 state $\mathbf{X}_1$ and cyclization state $\mathbf{x}_1^O$
2: $\widetilde{\mathbf{x}}^O \leftarrow \text{Copy}(\mathbf{x}_1^O)$     $\triangleright$ Initialize denoised cyclization-related atom coordinates $\widetilde{\mathbf{x}}^O$
3: $\mathcal{A}_1 \leftarrow \text{Uniform}(20, N)$     $\triangleright$ Randomly initialize residue types not constrained by cyclization
4: $\mathcal{G}_1 \leftarrow \text{Assemble}(\mathcal{A}_1, O)$     $\triangleright$ Derive initial chemical graph
5: $\mathbf{T} \leftarrow \mathbf{1}$     $\triangleright$ Initialize a timer that records time for each atom in atom73 state
6: $t \leftarrow 1$
7: **while** $t > \epsilon$ **do**
8:     $\mathbf{x}_t \leftarrow \text{Extract}(\mathbf{X}_t, \mathcal{A}_t) \cup \mathbf{x}_t^O$     $\triangleright$ Obtain all-atom $\mathbf{x}_t$ structure of current noisy peptide
9:     $\widehat{\mathbf{x}}_0 \leftarrow \text{ATOMSDE}(\mathbf{x}_t, \mathcal{G}_t, t)$     $\triangleright$ Predict denoised all-atom structure $\widehat{\mathbf{x}}_0$ structure
10:     $\widetilde{\mathbf{x}}^O \leftarrow \text{Cache}(\widetilde{\mathbf{x}}^O, \widehat{\mathbf{x}}_0, O)$
11:     $\mathbf{x}_{t-dt} \leftarrow \text{Noise}(\widehat{\mathbf{x}}_0, \mathcal{G}_t, t-dt)$     $\triangleright$ $\mathcal{G}_t$ is required by harmonic noise
12:     $\mathbf{X}_{t-dt} \leftarrow \text{Cache}(\mathbf{X}_t, \mathbf{x}_{t-dt}, \mathcal{A}_t)$     $\triangleright$ Update $\mathbf{X}_t$ by saving new structures to specific states according to $\mathcal{A}_t$
13:     $\mathbf{x}_{t-dt}^O \leftarrow \text{Cache}(\mathbf{x}_t^O, \mathbf{x}_{t-dt}, O)$
14:     $\mathbf{T} \leftarrow \text{UpdateTimer}(\mathbf{T}, \mathcal{A}_t, t-dt)$     $\triangleright$ Update the timer for the newly-updated atoms to the latest time
15:     $\widetilde{\mathcal{G}} \leftarrow \text{Subgraph}(\mathcal{G}_t, \mathcal{A}_t, O)$     $\triangleright$ Hide side chains of residues not constrained by cyclization
16:     $\mathcal{A}_{t-dt} \leftarrow \text{RESROUTER}(\widehat{\mathbf{x}}_0, \widetilde{\mathcal{G}}, t)$     $\triangleright$ Predict sequence based on the denoised structure $\widehat{\mathbf{x}}_0$
17:     $\mathcal{G}_{t-dt} \leftarrow \text{Assemble}(\mathcal{A}_{t-dt}, O)$     $\triangleright$ Derive a new chemical graph given predicted sequence and cyclization
18:     $\mathbf{t} \leftarrow \text{Extract}(\mathbf{T}, \mathcal{A}_{t-dt})$     $\triangleright$ Obtain atom-wise time $\mathbf{t}$ (Atom might have different time)
19:     $\mathbf{x}_{t-dt} \leftarrow \text{Extract}(\mathbf{X}_{t-dt}, \mathcal{A}_{t-dt}) \cup \mathbf{x}_{t-dt}^O$     $\triangleright$ Align atoms with different time $\mathbf{t}$ to the same time $t-dt$
20:     $\mathbf{x}_{t-dt} \leftarrow \text{AlignTime}(\mathbf{x}_{t-dt}, \widehat{\mathbf{x}}_0, \mathcal{G}_{t-dt}, t-dt, \mathbf{t})$     according to the reverse-time harmonic SDE
21:     $\mathbf{X}_{t-dt} \leftarrow \text{Cache}(\mathbf{X}_{t-dt}, \mathbf{x}_{t-dt}, \mathcal{A}_{t-dt})$     $\triangleright$ Store the time-aligned atom coordinates
22:     $t \leftarrow t - dt$
23: **end while**
24: $\mathbf{x}_t \leftarrow \text{Extract}(\mathbf{X}_t, \mathcal{A}_t) \cup \mathbf{x}_t^O$
25: $\mathbf{x}_0 \leftarrow \text{ATOMSDE}(\mathbf{x}_t, \mathcal{G}_t, t)$     $\triangleright$ Predict the all-atom structure finally
26: $\mathcal{G}_0 \leftarrow \mathcal{G}_t$
27: $\mathcal{A}_0 \leftarrow \mathcal{A}_t$

---

where $\beta(t)$ is a positive time-dependent scalar function, $\mathbf{P}$ is an orthogonal matrix (i.e., $\mathbf{P}\mathbf{P}^\mathsf{T} = \mathbf{I}$), $\boldsymbol{\Lambda} = \mathrm{diag}(\lambda_1, \dots, \lambda_{N_L})$ is a diagonal matrix that contains the eigenvalues, and $\mathbf{H} = \mathbf{P}\boldsymbol{\Lambda}\mathbf{P}^\mathsf{T}$.

We denote the variance of the random variable $\mathbf{x}^L$ as $\boldsymbol{\Sigma}(t)$, i.e., $\boldsymbol{\Sigma}(t) \coloneqq \mathrm{Cov}[\mathbf{x}(t)]$ for $t \in [0, 1]$. The aforementioned SDE, characterized by affine drift and diffusion coefficients, allows us to employ Eq. (5.51) from Särkkä & Solin (2019) to derive an ODE that describes the evolution of variance as follows:

$$\frac{\mathrm{d}\boldsymbol{\Sigma}}{\mathrm{d}t} = \beta(t)\big((\boldsymbol{\Lambda}^{\frac{1}{2}}\mathbf{P}^\mathsf{T})^\mathsf{T}\boldsymbol{\Lambda}^{\frac{1}{2}}\mathbf{P}^\mathsf{T} - \boldsymbol{\Sigma}(t)\big),$$
$$= \beta(t)(\mathbf{H} - \boldsymbol{\Sigma}(t)).$$

Solving the above ODE, we derive

$$\boldsymbol{\Sigma}(t) = \mathbf{H} + e^{\int_0^t -\beta(s)\mathrm{d}s}(\boldsymbol{\Sigma}(0) - \mathbf{H}),$$

Once the boundary condition $\mathbf{x}_0^L$ is given, we have $\boldsymbol{\Sigma}(0) = \mathbf{0}$. Thus, the induced perturbation kernel has an analytic form as:

$$p_{0t}(\mathbf{x}_t^L|\mathbf{x}_0^L) = \mathcal{N}(\mathbf{x}_t^L; \mathbf{x}_0^L e^{-\frac{1}{2}\int_0^t \beta(s)\mathrm{d}s}, \mathbf{H} - \mathbf{H}e^{-\int_0^t \beta(s)\mathrm{d}s}).$$

Given $\lim_{t \to 1} \int_0^t \beta(s)\mathrm{d}s = \infty$, the above perturbation process arrives at the prior distribution $p_1(\mathbf{x}_1^L) = \mathcal{N}(\mathbf{x}_1^L; \mathbf{0}, \mathbf{H})$.

# G    Implementation Details

## G.1    Model Architecture

Given a noisy sample at time $t$, two graphs are built for message passing with an SE(3)-equivariant neural network, which is parameterized by $\phi_K, \phi_B, \phi_C, \phi_H, \phi_E, \psi_K, \psi_C$ as introduced below. The $i$-th atom in the complex is attributed with an initial feature $\mathbf{h}_i$ and the bond $ij$ in the noisy ligand is attributed with an initial feature $\mathbf{b}_{ij}$. We first construct a k-nearest neighbor (knn) graph $\mathcal{G}_K$ for the complex (i.e., the protein and the noisy ligand at time $t$), where each ligand atom is connected with the k-nearest atoms in the complex, to capture the protein-ligand interaction:

$$\Delta\mathbf{h}_i^K \leftarrow \sum_{j \in \mathcal{N}_K(i)} \phi_K(\mathbf{h}_i, \mathbf{h}_j, \|\mathbf{x}_i - \mathbf{x}_j\|, E_{ij}, t),$$

where $\mathcal{N}_K(i)$ is the neighbors of atom $i$ in $\mathcal{G}_K$, $E_{ij}$ indicates the edge $ij$ is a protein-protein, ligand-ligand or protein-ligand edge.

We also leverage the chemical graph $\mathcal{G}_C$ of the ligand as we have defined previously to make the model aware of the connection information introduced by the chemical bonds:

$$\mathbf{e}_{ij} \leftarrow \phi_B(\|\mathbf{x}_i - \mathbf{x}_j, \mathbf{b}_{ij}\|),$$
$$\mathbf{h}_i^C \leftarrow \sum_{j \in \mathcal{N}_C(i)\phi_C}(\mathbf{h}_i, \mathbf{h}_j, \mathbf{e}_{ij}, t).$$

We further aggregate the hidden features of ligand atoms and bonds from these two graphs as follows:

$$\mathbf{h}_i \leftarrow \mathbf{h}_i + \phi_H(\Delta\mathbf{h}_i^K + \Delta\mathbf{h}_i^C),$$
$$\mathbf{b}_{ij} \leftarrow \sum_{k \in \mathcal{N}_C(j)\setminus\{i\}} \phi_B(\mathbf{h}_i, \mathbf{h}_j, \mathbf{h}_k, \mathbf{e}_{ik}, \mathbf{e}_{kj}, t).$$

Finally, we update the ligand atom positions as follows:

$$\Delta\mathbf{x}_i^K \leftarrow \sum_{j \in \mathcal{N}_K(i)} (\mathbf{x}_j - \mathbf{x}_i)\psi_K(\mathbf{h}_i, \mathbf{h}_j, \|\mathbf{x}_i - \mathbf{x}_j\|, t),$$
$$\Delta\mathbf{x}_i^C \leftarrow \sum_{j \in \mathcal{N}_C(i)} (\mathbf{x}_j - \mathbf{x}_i)\psi_K(\mathbf{h}_i, \mathbf{h}_j, \|\mathbf{x}_i - \mathbf{x}_j\|, \mathbf{e}_{ij}, t),$$
$$\mathbf{x}_i \leftarrow \mathbf{x}_i + (\Delta\mathbf{x}_i^K + \Delta\mathbf{x}_i^C) \cdot \mathbb{1}\{i \in \mathcal{G}_C\},$$

where $\mathbb{1}\{i \in \mathcal{G}_C\}$ indicates whether atom $i$ belongs to the ligand since the protein atom positions are fixed and we only update ligand atom positions.

We denote the final output of the SE(3)-equivariant neural network as $\boldsymbol{D}_{\boldsymbol{\theta}}(\mathbf{x}_t^L, t)$, where $\boldsymbol{D}_{\boldsymbol{\theta}}$ is composed of $\phi_K, \phi_B, \phi_C, \phi_H, \phi_E, \psi_K, \psi_C$ as introduced above.

### G.2 Training Details

We use the same optimizer setting for both ATOMSDE and RESROUTER: AdamW (Loshchilov, 2017) optimizer with constant learning rate 0.0001, beta1 0.9, beta2 0.999, and weight decay 0.01. For beta schedule, we use $\beta(t) = (\beta_{max} - \beta_{min})t + \beta_{min}$, where $\beta_{min} = 0.01$ and $\beta_{max} = 3.0$. Note that $\lim_{t \to 1} \int_0^s \beta(s)\mathrm{d}s$ is sufficiently large compared to the variance of our data distribution. To train ATOMSDE, we sample $t \sim \mathcal{U}[0, 1]$. To train RESROUTER, we sample $t \sim \mathcal{U}[0, 0.5]$ due to the fact that the denoised structure output by trained model ATOMSDE at time $t = 0.5$ or more has extremely limited information to determine the residue types. ATOMSDE converges within 48 hours and RESROUTER converges within 18 hours on 8 NVIDIA H100 GPUs.

### G.3 Sampling Details

For routed sampling, we divide time interval $[0, 1]$ into 1,000 steps. Inspired by Chu et al. (2024), we skip RESROUTER when $t > 0.5$ since the structures are too noisy to provide sufficient information for residue type prediction. This approach also accelerates the generative process and reduces the computational cost of inference. When $t < 0.5$, at each step, sequence and structures are iteratively updated by RESROUTER and ATOMSDE, respectively.

## H Experimental Details

### H.1 Relaxation and Energy Estimation

Cyclic peptides offer notable advantages in terms of both system stability and binding affinity. In specific, the stability of a protein-peptide complex is inversely proportional to its overall free energy, with lower free energy indicating greater stability. To assess this, the `FastRelax` protocol in PyRosetta (Chaudhury et al., 2010) is employed to relax each complex, after which the total energy is evaluated using the REF2015 scoring function. Binding affinity is measured with the `InterfaceAnalyzerMover`, which calculates the binding energy at the interface between the peptide and the target protein within the relaxed complex. An increase in binding energy reflects enhanced peptide binding affinity, suggesting potential functional improvements.

For each target, linear peptide methods generate 8 samples with the golden peptide length (i.e., the number of residues within reference linear peptide). Cyclic peptide methods, lacking a reference length, enumerate residues from 5 to 20 (or 8 to 23 for side-to-side cyclic peptides), generating 2 samples per length. All reference ligands and samples are relaxed and evaluated using a standard scoring method as described above. For each target and method, we apply the Borda method to select the best ligand, accounting for both stability and affinity, which is then reported in the final results.

### H.2 Detailed Experimental Results

We have provided detailed energy measurement results for each target, including the reference ligand, linear peptides designed by baselines, and cyclic peptides designed by our methods, in Tables 6 and 7.

### H.3 Inference Speed

We benchmark the average time of generating one peptide for all co-design baselines and our methods on a single NVIDIA A100-SXM4-80GB GPU. The results are reported in Table 2. Given that computational drug design does not demand real-time model response, the inference time of our method is deemed acceptable.

Table 2. Generation time of all co-design baselines and our methods.

| Method | Peptide Type | Time (s) |
|---|---|---|
| ProteinGenerator | Linear | 31.80 |
| PepFlow | Linear | 12.09 |
| PepGLAD | Linear | 4.40 |
| CPSDE | Cyclic | 16.88 |

### H.4 Evaluation on Linear Peptide Design

While designing linear peptides is not our primary focus, we compare our method with existing baselines in this task.

Unlike the variable-length setting for cyclic peptides, the task of linear peptide design can leverage known reference peptide lengths for target proteins in the test set. For fair comparison across methods, we sample 8 linear peptides per target matching the reference length and relax the complex structure by Rosetta (Chaudhury et al., 2010; Alford et al., 2017). We then apply the Borda method to choose the optimal linear peptides, considering both stability and affinity. We report the average and median for Stability and Affinity of the linear peptides for targets in the test set. We also report the average for the fraction of hydrophobic and charged residues (relevant for specificity) (Ye et al., 2024), DockQ, iRMSD, LRMSD, BSR, and Diversity.

We use DockQ package [4] to compute DockQ, iRMSD, and LRMSD. We follow the definition of binding site ratio (BSR) in Li et al. (2025). A lower hydrophobic/charged ratio indicates a lower risk of non-specific binding (Makowski et al., 2024). The results are reported in Table 3. Notably, the fraction of hydrophobic and charged residues of our designed peptides resembles that of reference. Our method also shows superiority in structural properties.

*Table 3.* Summary of properties of reference peptides, linear peptides designed by baseline methods and CPSDE. (↓) / (↑) denotes a smaller / larger number is better.

| Method | Co-Design | Peptide Type | Stability (↓) Avg. | Stability (↓) Med. | Affinity (↓) Avg. | Affinity (↓) Med. | Hydrophobic Ratio (↓) | Charged Ratio (↓) | DockQ (↑) | iRMSD (↓) | LRMSD (↓) | BSR (↑) | Diversity (↑) |
|---|---|---|---|---|---|---|---|---|---|---|---|---|---|
| Reference | N/A | Linear | -672.53 | -634.71 | -85.03 | -78.70 | 0.48 | 0.28 | N/A | N/A | N/A | N/A | N/A |
| RFDiffusion | ✗ | Linear | -633.51 | -607.82 | -70.30 | -61.35 | 0.59 | 0.27 | 0.18 | 5.37 | 20.10 | 0.33 | 0.55 |
| ProteinGenerator | ✓ | Linear | -576.39 | -554.70 | -46.98 | -40.39 | 0.53 | 0.32 | 0.12 | 5.56 | 23.97 | 0.20 | 0.58 |
| PepFlow | ✓ | Linear | -576.16 | -498.31 | -47.88 | -42.40 | 0.60 | 0.17 | 0.44 | 2.49 | 9.42 | 0.56 | 0.70 |
| PepGLAD | ✓ | Linear | -359.44 | -310.33 | -45.06 | -38.56 | 0.53 | 0.25 | 0.30 | 2.68 | 11.99 | 0.39 | 0.79 |
| CPSDE | ✓ | Linear | -567.34 | -510.58 | -55.48 | -49.89 | 0.45 | 0.24 | 0.32 | 2.36 | 9.91 | 0.60 | 0.77 |

## H.5 Ablation Studies

**Effects of RESROUTER.** We study the effects of RESROUTER compared with the following two setups: "w/ fix seq" where the residue types are randomly sampled and fixed with only atom coordinates updated during the generative process, "w/ random seq" where the residue types are randomly sampled from a uniform distribution instead of predicted by RESROUTER during the generative process. The results are shown in Table 4. It can be observed that both variants perform worse than CPSDE, which demonstrates that RESROUTER can effectively discover critical residue types for protein-ligand interaction. "w/ random seq" performs the worst, possibly because random residue types offer no information gain, and the residue types are updated too frequently. This frequent updating hinders the ATOMSDE from effectively updating the side-chain atoms of the free residues.

*Table 4.* Ablation study on the effect of RESROUTER.

| Method | Stability (↓) Avg. | Stability (↓) Med. | Affinity (↓) Avg. | Affinity (↓) Med. |
|---|---|---|---|---|
| CPSDE | -568.04 | -519.66 | -50.86 | -46.62 |
| w/ fix seq | -525.73 | -439.69 | -41.56 | -38.75 |
| w/ random seq | -521.48 | -425.17 | -38.64 | -39.44 |

**Effects of Harmonic SDE.** We study the effects of harmonic SDE. We introduce a variant with isotropic Gaussian as prior and noise distribution (denoted as "w/o Harmonic"). The results are shown in Table 4 and validate the effectiveness of harmonic prior and noise. To explore the underlying reasons, we examined the trajectories of the generative process and found that the harmonic prior provides a good initialization for atom positions, where bonded atoms are located nearby. This feature might be beneficial for routed sampling because some side-chain atom updates can be discontinuous, and such correlated initialization helps mitigate the errors induced by these discontinuous updates.

*Table 5.* Ablation study on the effect of Harmonic SDE.

| Method | Stability (↓) Avg. | Stability (↓) Med. | Affinity (↓) Avg. | Affinity (↓) Med. |
|---|---|---|---|---|
| CPSDE | -568.04 | -519.66 | -50.86 | -46.62 |
| w/o Harmonic | -534.38 | -439.23 | -39.43 | -41.08 |

## H.6 System Setup and Protocols of Molecular Dynamics Simulation

To simulate the protein-peptide systems, hydrogen atoms are added, and the dominant protonation state of titratable residues at pH 7 is determined using PropKa in PDB2PQR (Dolinsky et al., 2007). Subsequently, the systems are solvated in a 10 Å truncated water box, with sodium and chloride ions added to neutralize the system at a concentration of 150 mM to mimic physiological saline. The ff14SB (Maier et al., 2015) parameter set is applied to proteins and peptides, and the TIP3P model is used for water (Jorgensen et al., 1983; Li et al., 2024).

All simulations were run on RTX 4090 GPUs using the CUDA implementation of particle-mesh Ewald (PME) molecular

---

[4] https://github.com/bjornwallner/DockQ

dynamics in Amber22 (Salomon-Ferrer et al., 2013). At first, to relax each system thoroughly, two stages of energy minimization are performed. In the first stage, 2,500 steepest descent and 2,500 conjugate gradient cycles were applied to all atoms, with constraints on water molecules and counterions. In the second stage, the same cycles were repeated without constraints. Initial velocities are randomly sampled from a Boltzmann distribution. The systems are then heated from 0 K to 310 K over 500 ps in the NVT ensemble, using a Langevin thermostat and harmonic restraints of $10.0 \text{ kcal} \cdot \text{mol}^{-1} \cdot \text{Å}^{-2}$. During equilibration at 300 K and 1 bar under NPT conditions, harmonic restraints on protein and peptide atoms were progressively reduced from 5.0 to $0.1 \text{ kcal} \cdot \text{mol}^{-1} \cdot \text{Å}^{-2}$ in four steps at 0.5 ns intervals, totaling 2.5 ns. All restraints are completely removed during production simulation under 310K and 1 bar, which are maintained using the Langevin thermostat and Berendsen barostat, respectively. A timestep of 4.0 fs is used with hydrogen mass repartitioning (Hopkins et al., 2015). Bond lengths are constrained via SHAKE (Ryckaert et al., 1977), and non-bonded interactions are cut off at 10 Å.

### H.7 Visualization of Structure Ensembles Simulated by Molecular Dynamics

We present additional views of the structure ensembles generated by molecular dynamics simulations in Figure 12 and Figure 13.

## I Examples of Designed Cyclic Peptides

Here, we present more compelling results of our generated cyclic peptides targeting different receptors in Figures 9, 14 and 15. We find that our generated cyclic peptides consistently exhibit higher or competitive affinities with greater interaction stabilities when binding to the receptor. In contrast, linear peptides sampled from PepFlow often result in unstable structures and weaker binding. Furthermore, our designed cyclic peptides not only interact with key receptor regions, similar to linear peptides and native peptides, but also establish new, stable, and tight interactions in additional regions. Additionally, our generated 3D cyclic peptides consistently align well with the corresponding 2D chemical graphs, highlighting the effective integration of our two models.

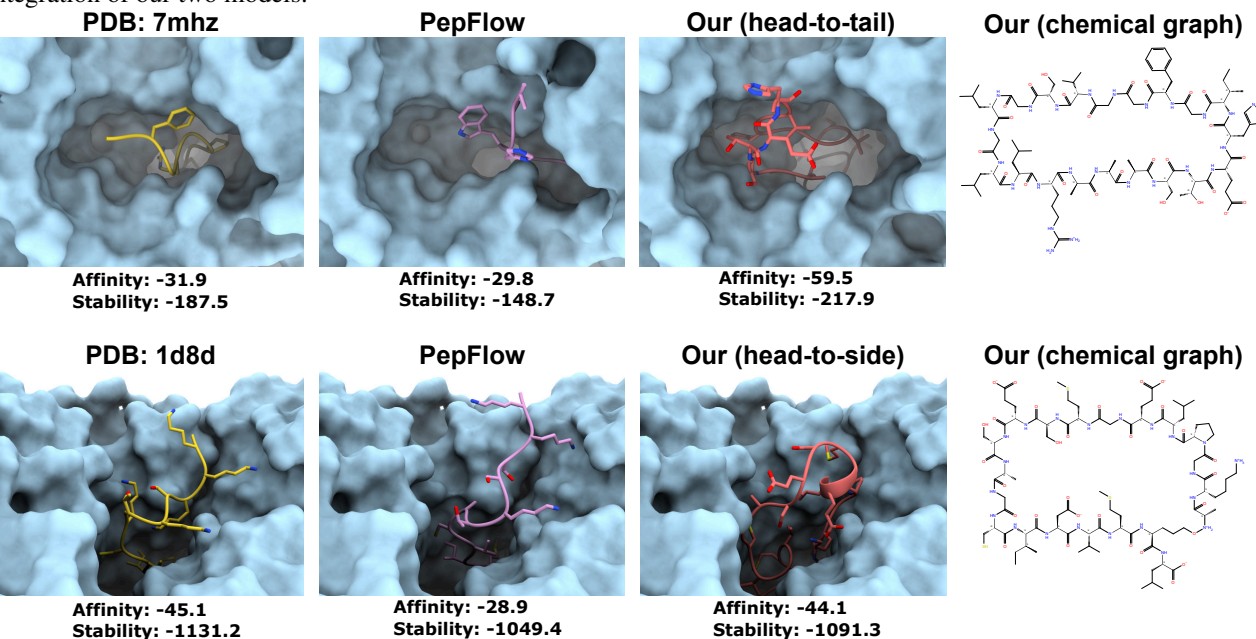

*Figure 9.* Visualization of reference ligand, linear peptides designed by PepFlow, and cyclic peptides designed by CPSDE.

## J Limitations and Future Work

One limitation is that the generated cyclic peptides may sometimes exhibit invalid conformations, such as inaccurate bond lengths and atomic receptor clashes. While Rosetta-based structure relaxation (Chaudhury et al., 2010; Alford et al., 2017) can refine these structures, it is computationally expensive and slow. Additionally, for evaluation, we are currently unable to introduce self-consistency metrics similar to those in protein design (Yim et al., 2024), as there is no highly accurate cyclic peptide structure prediction or docking model available.

*Table 6.* Stability and affinity of the reference peptide, linear peptides designed by baseline methods, and cyclic peptide designed by our method along with the cyclization type.

| Target | Reference | | RFDiffusion | | ProteinGenerator | | PepFlow | | PepGLAD | | Our | | |
|---|---|---|---|---|---|---|---|---|---|---|---|---|---|
| | Stab. | Affi. | Stab. | Affi. | Stab. | Affi. | Stab. | Affi. | Stab. | Affi. | Stab. | Affi. | Type |
| 1d8d | -1131.2 | -45.1 | -1092.2 | -37.2 | -975.5 | -43.7 | -1049.4 | -28.9 | -1070.0 | -30.1 | -1091.3 | -44.1 | h2s |
| 1hr8 | -417.7 | -48.4 | -363.9 | -78.7 | -370.0 | -39.3 | -428.8 | -36.8 | -207.8 | -28.5 | -313.1 | -37.2 | s2t |
| 1rgq | -374.0 | -140.2 | -390.1 | -153.6 | -289.1 | -121.1 | -167.5 | -69.3 | 82.6 | -67.2 | -255.1 | -83.5 | s2t |
| 1vzj | -80.1 | -138.7 | 282.8 | -66.3 | 356.1 | -52.8 | 12.4 | -112.2 | 20.4 | -90.6 | 143.4 | -105.3 | h2s |
| 1xoc | -899.3 | -66.9 | -763.8 | -40.7 | -866.0 | -41.2 | -886.2 | -51.1 | -778.8 | -30.8 | -867.3 | -47.7 | h2s |
| 1zkk | -824.2 | -47.0 | -843.0 | -59.0 | -824.0 | -40.2 | -797.3 | -30.0 | -439.0 | -35.5 | -793.2 | -62.9 | h2t |
| 2arq | -556.3 | -148.5 | -588.7 | -165.9 | -428.5 | -29.2 | -428.2 | -63.5 | -204.1 | -53.1 | -438.3 | -91.5 | h2t |
| 2mpz | -1136.2 | -223.1 | -1114.8 | -203.7 | -1031.2 | -91.4 | -937.5 | -84.0 | 491.4 | -79.3 | -1059.0 | -244.1 | h2s |
| 2vda | 223.6 | -59.6 | 336.7 | -33.5 | 393.3 | -32.4 | 343.5 | -35.1 | 960.0 | -47.8 | 373.1 | -25.9 | h2t |
| 2wqj | -576.8 | -123.3 | -571.3 | -125.7 | N/A | N/A | -447.3 | -71.8 | 126.2 | -76.0 | -386.2 | -48.8 | h2t |
| 2xjz | 95.0 | -164.0 | 234.9 | -102.2 | 343.4 | -58.1 | 251.8 | -72.0 | N/A | N/A | 375.2 | -46.6 | h2t |
| 3e8e | -1144.4 | -53.3 | -1123.9 | -50.8 | -1119.1 | -39.1 | -1020.6 | -20.3 | -551.5 | -27.0 | -1081.7 | -37.4 | h2t |
| 3ech | -569.4 | -101.1 | -541.2 | -78.8 | -476.6 | -84.8 | -446.8 | -44.0 | -231.2 | -65.0 | -436.8 | -56.0 | h2t |
| 3ewf | -1453.3 | -36.4 | -1425.1 | -16.9 | -1433.9 | -11.7 | -1356.4 | 0.4 | -1390.7 | -11.9 | -1484.1 | -48.0 | h2s |
| 3fii | -290.1 | -105.2 | -197.1 | -88.8 | -173.3 | -34.7 | -29.4 | -30.5 | -39.3 | -100.2 | -100.3 | -45.9 | h2t |
| 3h8a | -528.3 | -71.1 | -538.8 | -68.6 | -446.2 | -57.7 | -400.1 | -58.1 | 232.1 | -38.3 | -442.8 | -45.2 | h2s |
| 3j89 | -919.6 | -118.1 | -908.3 | -115.3 | -876.1 | -84.7 | -771.6 | -59.9 | N/A | N/A | -744.2 | -56.5 | h2t |
| 3lk4 | -263.2 | -101.4 | -215.7 | -80.6 | -176.2 | -26.4 | -256.9 | -68.3 | N/A | N/A | -223.3 | -31.5 | h2s |
| 3mhp | -1021.0 | -93.4 | -948.3 | -64.0 | -851.9 | -13.8 | -903.4 | -43.8 | -682.8 | -59.5 | -849.0 | -38.0 | h2t |
| 3o0e | -353.1 | -43.0 | -286.4 | -55.7 | -338.0 | -42.9 | -357.9 | -33.1 | -126.2 | -30.8 | -332.9 | -30.4 | s2t |
| 3pl7 | -215.7 | -105.9 | -266.8 | -124.2 | -257.3 | -104.2 | -140.4 | -98.1 | 398.6 | -98.9 | -81.0 | -59.1 | h2s |
| 3ro2 | -606.5 | -86.2 | -376.7 | -32.1 | -345.0 | -31.6 | -539.1 | -52.5 | -117.2 | -45.8 | -378.1 | -42.9 | h2s |
| 3ryb | -990.0 | -54.9 | -993.3 | -59.4 | -894.1 | -34.5 | -1003.8 | -48.2 | -857.7 | -37.0 | -942.6 | -47.4 | h2t |
| 3twt | -965.3 | -57.3 | -862.1 | -23.2 | -851.4 | -14.4 | -936.2 | -31.5 | -782.6 | -31.8 | -892.3 | -36.6 | h2s |
| 3vvs | -691.1 | -55.1 | -632.5 | -40.1 | -634.7 | -49.6 | -692.0 | -50.3 | -632.0 | -38.1 | -722.5 | -63.7 | h2s |
| 3wy9 | -618.2 | -59.8 | -522.1 | -73.3 | -610.9 | -56.6 | -523.7 | -34.0 | -313.7 | -49.7 | -550.8 | -24.4 | h2t |
| 3zha | -404.5 | -122.1 | -187.8 | -37.2 | -136.9 | -62.0 | -308.1 | -60.1 | -451.7 | -68.2 | -261.4 | -55.9 | h2s |
| 4chg | -936.0 | -112.9 | -790.8 | -59.4 | -809.6 | -77.7 | -790.1 | -66.7 | N/A | N/A | -714.0 | -64.2 | h2t |
| 4e7v | -490.4 | -125.4 | N/A | N/A | N/A | N/A | -308.1 | -73.0 | 53.7 | -96.0 | -473.1 | -158.2 | h2t |
| 4edn | -800.6 | -75.1 | -815.0 | -75.6 | -694.9 | -24.8 | -760.6 | -44.3 | -521.4 | -35.6 | -711.4 | -53.3 | h2t |
| 4hom | -1143.1 | -33.5 | -1108.8 | -56.5 | -1190.9 | -22.5 | -1108.6 | -36.9 | -1105.9 | -21.9 | -1184.8 | -34.2 | h2t |
| 4jo6 | -925.4 | -81.9 | -927.8 | -72.0 | -861.7 | -40.6 | -792.0 | -32.6 | -131.5 | -60.4 | -759.1 | -45.4 | s2s |
| 4m1c | -156.4 | -41.1 | -98.9 | -26.8 | -16.6 | -28.4 | -100.1 | -33.6 | -74.8 | -25.5 | -102.3 | -30.9 | h2s |
| 4o6f | -439.3 | -49.9 | -422.1 | -66.9 | -425.3 | -49.2 | -428.8 | -32.3 | -370.9 | -21.7 | -425.6 | -44.1 | h2t |
| 4po7 | -301.2 | -42.9 | -294.3 | -65.0 | -244.1 | -56.2 | -294.1 | -25.4 | -267.9 | -20.9 | -261.9 | -24.9 | h2t |
| 4qae | -912.4 | -94.9 | -945.3 | -67.4 | -927.2 | -46.4 | -870.7 | -56.3 | -198.1 | -55.6 | -851.1 | -54.3 | s2s |
| 4uqz | -733.4 | -96.8 | -652.8 | -30.9 | -640.8 | -35.5 | -557.0 | -39.6 | -555.3 | -38.1 | -628.2 | -48.0 | s2s |
| 4wsi | -281.4 | -93.5 | -145.6 | -58.1 | -107.4 | -34.6 | -180.7 | -30.9 | -85.3 | -45.6 | -210.8 | -32.0 | h2t |
| 4x3o | -495.0 | -28.0 | -494.6 | -30.6 | -420.4 | -21.3 | -441.6 | 0.0 | -468.4 | -16.8 | -585.4 | -83.4 | h2t |
| 4xpd | -114.1 | -23.3 | -62.1 | -20.3 | -78.3 | -30.0 | -27.9 | -6.3 | 0.3 | -15.6 | -175.7 | -48.9 | h2t |
| 4xtr | -660.2 | -110.7 | -659.0 | -97.4 | -662.6 | -77.0 | -594.0 | -81.9 | -372.5 | -57.4 | -534.9 | -62.6 | h2s |
| 4yjl | -1013.7 | -73.3 | -859.8 | -39.6 | -814.8 | -24.0 | -934.8 | -38.7 | -745.0 | -31.4 | -923.3 | -30.8 | h2t |
| 4zp3 | -384.2 | -102.1 | -414.8 | -119.2 | -228.4 | -42.2 | -228.8 | -61.0 | 24.9 | -71.6 | -214.1 | -39.5 | h2s |
| 5apk | -367.8 | -62.1 | -240.0 | -65.1 | -146.3 | -62.7 | -310.0 | -48.0 | -269.8 | -33.2 | -330.1 | -60.7 | h2t |
| 5brm | -469.6 | -72.2 | -379.6 | -38.4 | -375.3 | -24.4 | -405.0 | -42.6 | -301.6 | -44.9 | -414.8 | -43.8 | h2t |
| 5c6h | -358.6 | -83.3 | -350.2 | -87.3 | -220.6 | -36.3 | -301.3 | -92.8 | -29.3 | -65.9 | -231.5 | -46.6 | h2t |
| 5dhm | -589.5 | -155.4 | -607.8 | -149.4 | -451.7 | -64.2 | -483.1 | -39.4 | -123.1 | -77.9 | -424.9 | -53.1 | h2t |
| 5e2q | -977.0 | -65.2 | -865.6 | -37.4 | -839.6 | -28.8 | -932.5 | -47.5 | -819.0 | -31.9 | -902.4 | -58.5 | s2s |
| 5et1 | -1436.7 | -77.3 | -1321.0 | -41.4 | -1391.5 | -54.9 | -1318.7 | -32.9 | -592.4 | -37.1 | -1313.0 | -26.5 | h2t |
| 5iyx | -744.4 | -59.5 | -612.4 | -7.7 | -652.9 | -37.8 | -659.4 | -34.7 | -564.2 | -28.3 | -653.3 | -38.1 | h2t |
| 5j3t | -718.3 | -102.9 | -624.8 | -57.4 | -586.4 | -33.8 | -509.8 | -27.3 | -211.8 | -80.0 | -501.0 | -36.7 | h2t |
| 5mfg | -1215.8 | -42.3 | -1171.4 | -28.1 | -1195.7 | -15.5 | -1236.5 | -42.2 | -1049.8 | -36.3 | -1214.7 | -40.0 | h2s |
| 5mjy | -1527.3 | -79.4 | -1538.3 | -68.7 | -1369.1 | -48.1 | -1482.5 | -58.8 | -1299.2 | -43.8 | -1425.3 | -45.5 | h2s |
| 5n4d | -1231.2 | -61.8 | -1206.1 | -62.6 | -1189.6 | -39.4 | -1219.5 | -73.4 | -1034.4 | -35.4 | -1184.8 | -40.9 | s2s |
| 5nl1 | -669.8 | -79.2 | -677.0 | -87.6 | -702.1 | -91.6 | -572.6 | -69.0 | -354.6 | -56.7 | -532.3 | -44.3 | s2s |
| 5txe | -955.6 | -49.6 | -930.2 | -48.3 | -892.9 | -48.2 | -883.6 | -39.8 | -860.2 | -38.9 | -879.2 | -41.1 | h2t |
| 5vt9 | -622.4 | -128.5 | -546.2 | -97.0 | -520.6 | -72.4 | -469.3 | -86.0 | 1.8 | -58.2 | -438.5 | -51.6 | h2s |
| 5wkf | 258.4 | -72.2 | 343.2 | -36.4 | 384.2 | -25.3 | 322.4 | -42.8 | 408.0 | -38.8 | 303.2 | -54.4 | s2t |
| 5wpl | -352.8 | -99.6 | -380.7 | -91.3 | -284.4 | -59.4 | -216.6 | -69.4 | N/A | N/A | -197.3 | -51.4 | s2s |
| 5yis | -416.4 | -99.1 | -291.1 | -34.1 | -263.1 | -57.8 | -265.5 | -50.1 | -149.5 | -50.0 | -302.1 | -56.1 | s2s |

*Table 7.* Stability and affinity of the reference peptide, linear peptides designed by baseline methods, and cyclic peptide designed by our method along with the cyclization type.

| Target | Reference | | RFDiffusion | | ProteinGenerator | | PepFlow | | PepGLAD | | Our | | |
|---|---|---|---|---|---|---|---|---|---|---|---|---|---|
| | Stab. | Affi. | Stab. | Affi. | Stab. | Affi. | Stab. | Affi. | Stab. | Affi. | Stab. | Affi. | Type |
| 5zw6 | -361.4 | -46.9 | -298.0 | -14.3 | -497.5 | -29.3 | -338.0 | -41.4 | -306.9 | -28.5 | -364.7 | -51.5 | h2s |
| 6bli | -1269.1 | -104.4 | -1231.7 | -62.2 | -1231.4 | -46.5 | -1147.9 | -48.4 | N/A | N/A | -1094.2 | -39.5 | h2s |
| 6cv1 | -490.1 | -109.1 | -425.7 | -131.3 | -284.3 | -44.2 | -254.6 | -38.5 | N/A | N/A | -234.9 | -36.1 | h2t |
| 6di8 | -1850.6 | -73.9 | -1805.2 | -49.3 | -1795.9 | -44.0 | -1783.6 | -52.4 | -1364.2 | -37.5 | -1793.5 | -59.9 | h2t |
| 6dtg | -874.6 | -60.3 | -792.4 | -19.5 | -698.6 | -27.9 | -847.4 | -32.4 | -783.4 | -22.8 | -898.9 | -52.4 | h2s |
| 6f0h | -665.8 | -80.0 | -499.4 | -57.0 | -544.4 | -46.0 | -529.0 | -32.7 | -280.6 | -54.8 | -539.4 | -38.9 | h2s |
| 6f6d | -771.9 | -88.3 | -666.2 | -47.1 | -653.4 | -63.8 | -674.9 | -39.8 | -599.7 | -25.0 | -666.4 | -39.3 | h2t |
| 6g68 | -303.2 | -121.6 | -349.0 | -128.1 | -352.1 | -126.1 | -148.8 | -83.8 | N/A | N/A | -15.7 | -57.9 | h2t |
| 6ghr | -1342.1 | -56.9 | -1198.8 | -70.7 | -1260.8 | -49.8 | -1212.9 | -30.0 | -666.1 | -43.8 | -1224.1 | -50.6 | h2t |
| 6ict | -1323.9 | -87.2 | -1259.5 | -37.0 | -1222.3 | -29.4 | -1142.0 | -26.6 | -662.6 | -36.7 | -1238.7 | -57.0 | h2s |
| 6igk | -785.6 | -104.9 | -768.3 | -84.0 | -762.4 | -64.6 | -698.1 | -75.3 | -472.5 | -52.7 | -680.6 | -62.2 | h2t |
| 6jbk | -938.6 | -79.7 | -933.7 | -76.5 | -881.2 | -42.2 | -871.9 | -50.2 | -375.9 | -63.7 | -847.5 | -40.5 | h2s |
| 6ocp | -754.5 | -51.1 | -672.6 | -15.4 | -565.0 | -31.0 | -716.6 | -30.7 | -552.5 | -22.7 | -676.5 | -34.2 | h2s |
| 6om4 | -690.0 | -67.6 | -582.3 | -49.2 | -506.8 | -9.4 | -662.7 | -25.1 | -485.6 | -22.6 | -710.9 | -47.6 | h2s |
| 6p02 | -1134.3 | -202.1 | -1136.5 | -171.3 | -1059.6 | -150.6 | -861.6 | -83.6 | -679.6 | -81.1 | -854.0 | -94.5 | h2s |
| 6peu | -893.2 | -78.2 | -832.3 | -46.2 | -736.2 | -7.2 | -935.1 | -61.8 | -807.9 | -22.6 | -860.5 | -44.7 | s2s |
| 6q5r | -529.4 | -94.9 | -585.9 | -132.9 | -566.2 | -109.0 | -437.5 | -98.9 | N/A | N/A | -401.3 | -60.9 | s2t |
| 6qs1 | -917.2 | -45.4 | -847.3 | -65.4 | -871.5 | -36.2 | -893.4 | -38.9 | -787.7 | -23.0 | -903.7 | -52.6 | h2s |
| 6r16 | -1330.4 | -102.6 | -1309.5 | -82.5 | -1263.7 | -47.8 | -1196.3 | -50.6 | -440.0 | -64.7 | -1181.7 | -50.5 | h2t |
| 6rqx | -1176.7 | -28.1 | -1069.6 | -32.2 | -1078.0 | -19.0 | -1151.5 | -20.3 | -1100.8 | -15.3 | -1207.8 | -37.6 | s2s |
| 6rxr | -449.0 | -66.1 | -472.8 | -49.5 | -427.0 | -30.4 | -466.3 | -35.8 | -80.7 | -42.2 | -499.6 | -192.0 | s2t |
| 6sa8 | -461.4 | -62.0 | -307.1 | -44.0 | -364.0 | -25.4 | -404.2 | -35.4 | -247.5 | -27.8 | -424.9 | -33.1 | h2s |
| 6trw | -1085.1 | -44.4 | -1074.5 | -48.0 | -947.0 | -40.2 | -1093.9 | -36.1 | -544.2 | -37.3 | -1080.4 | -48.9 | h2t |
| 6y1a | -393.6 | -202.3 | -358.1 | -191.0 | 39.8 | -49.7 | -153.8 | -84.7 | 264.3 | -68.4 | -142.2 | -69.2 | h2t |
| 6zw0 | -589.9 | -105.7 | -576.7 | -115.6 | -405.2 | -18.5 | -458.6 | -61.4 | -294.6 | -38.9 | -426.0 | -50.5 | h2s |
| 7atr | -1056.1 | -66.1 | -912.3 | -35.8 | -1036.8 | -33.5 | -991.2 | -37.1 | -978.0 | -33.5 | -999.1 | -49.0 | h2t |
| 7brk | -561.5 | -89.4 | -594.4 | -91.7 | -614.1 | -91.1 | -486.8 | -47.7 | -345.7 | -49.7 | -487.3 | -52.1 | h2s |
| 7eib | -342.6 | -50.3 | N/A | N/A | -331.3 | -45.8 | -276.4 | -34.4 | -270.9 | -37.7 | -387.1 | -63.7 | h2s |
| 7f6h | -238.6 | -50.3 | N/A | N/A | -149.4 | -24.6 | -186.3 | -38.7 | -185.7 | -35.7 | -271.7 | -57.3 | h2s |
| 7mhz | -187.5 | -31.9 | N/A | N/A | -74.8 | -27.0 | -148.7 | -29.8 | -48.7 | -34.2 | -217.9 | -59.5 | h2t |
| 7okp | -845.2 | -36.8 | N/A | N/A | -721.8 | -20.4 | -793.1 | -27.6 | -522.7 | -17.2 | -857.3 | -36.6 | h2t |
| 7owu | -469.1 | -50.8 | N/A | N/A | -454.9 | -39.1 | -464.5 | -23.4 | -293.0 | -27.0 | -487.5 | -62.8 | s2t |
| 7q66 | -375.5 | -189.6 | N/A | N/A | -306.4 | -130.5 | -17.6 | -88.6 | 241.7 | -54.4 | -96.6 | -80.0 | h2s |
| 7ure | -41.8 | -50.4 | -65.1 | -61.4 | 101.5 | -40.2 | 120.6 | -23.2 | 114.4 | -29.2 | 56.4 | -40.7 | h2s |
| 7vb7 | -156.8 | -92.4 | 62.6 | -78.2 | -23.8 | -33.1 | 47.1 | -38.2 | N/A | N/A | 19.1 | -20.6 | h2s |
| 7vwo | -647.0 | -93.5 | -519.6 | -58.2 | -435.8 | -41.1 | -464.2 | -44.9 | -125.6 | -72.4 | -523.5 | -54.8 | h2t |
| 7wvx | -314.7 | -81.9 | -170.0 | -54.2 | -221.3 | -45.4 | -159.7 | -40.5 | -177.8 | -58.6 | -254.4 | -64.9 | h2t |
| 7xxf | -243.6 | -73.7 | -281.3 | -71.4 | -259.1 | -71.7 | -154.1 | -54.7 | -64.0 | -54.4 | -115.9 | -44.2 | h2s |
| 7yat | -655.4 | -247.2 | -632.5 | -233.3 | -234.9 | -39.4 | -356.3 | -82.7 | 859.3 | -65.8 | -532.1 | -241.8 | h2s |
| 8dgq | -843.5 | -91.9 | -760.5 | -34.4 | -654.9 | -23.8 | -621.5 | -18.1 | -651.9 | -41.0 | -710.6 | -40.4 | h2t |

Property-guided sampling can be incorporated during generation to generate chemically and structurally valid cyclic peptides (Dhariwal & Nichol, 2021; Ho & Salimans, 2022). For example, the process can be conditioned on predefined bond length and angle distributions to sample cyclic peptides with specific shapes. Additionally, energy-based sampling (Lu et al., 2023; Kulytė et al., 2024) and energy-based preference optimization (Zhou et al., 2024a;c;d; Cheng et al., 2024) can guide the generation of low-energy, stable conformations. Techniques and architectures related to AlphaFold 3 could also be leveraged for more accurate atomic interaction modeling (Abramson et al., 2024). For evaluation, we believe it is crucial to validate the cyclic peptides generated by our model in the wet lab, determining their accurate structural conformations and binding modes—an avenue we are actively exploring.

Additionally, we would like to point out that our current method does not explore how to automatically select the best cyclization type for a given receptor, although it can be enumerated. We plan to investigate this aspect in our future work. Other future research includes cyclic ligand peptide design considering flexible protein targets (Zhou et al., 2025) or dual targets (Zhou et al., 2024b).

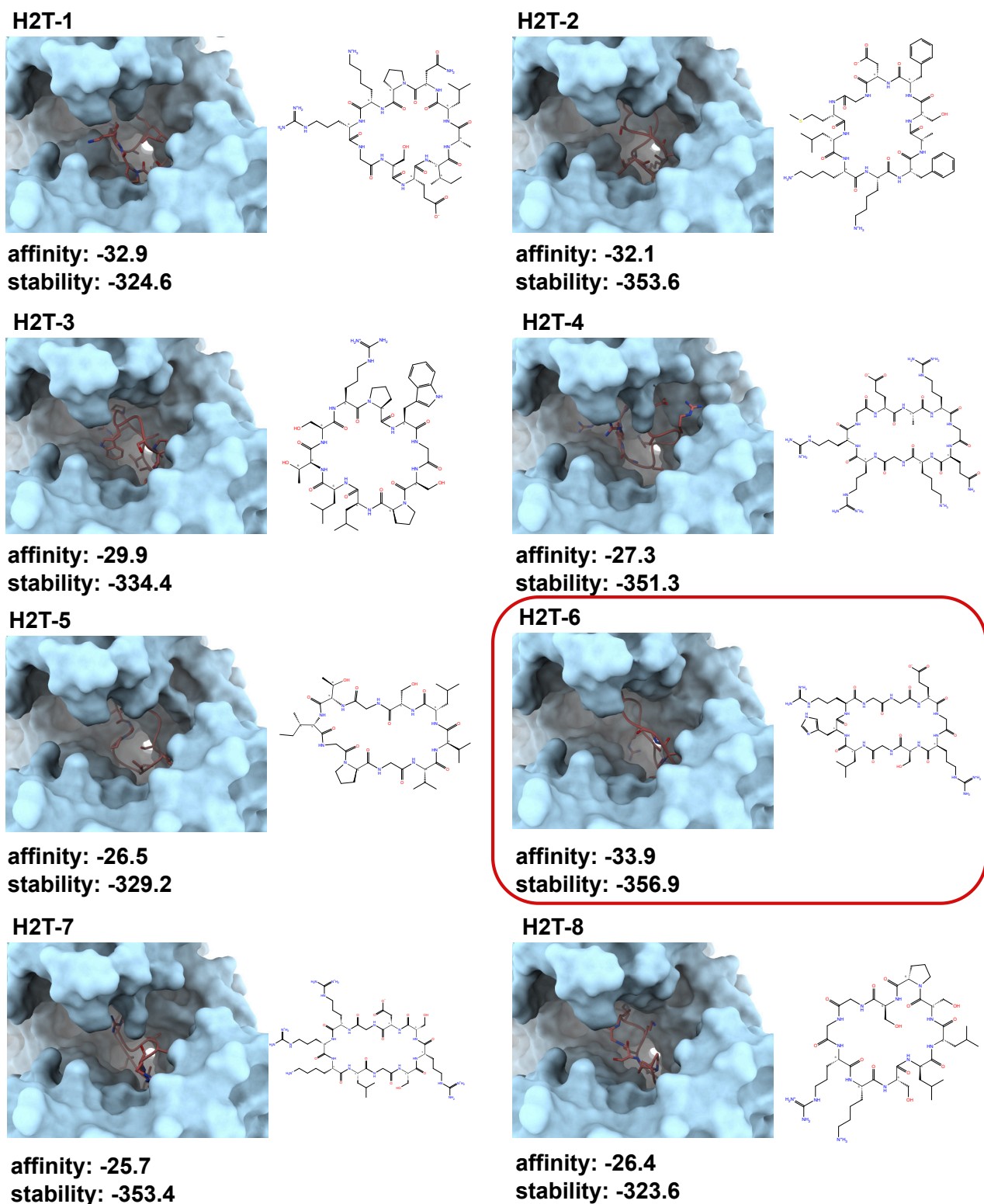

*Figure 10.* Head-to-tail cyclic peptides designed for target SMYD2.

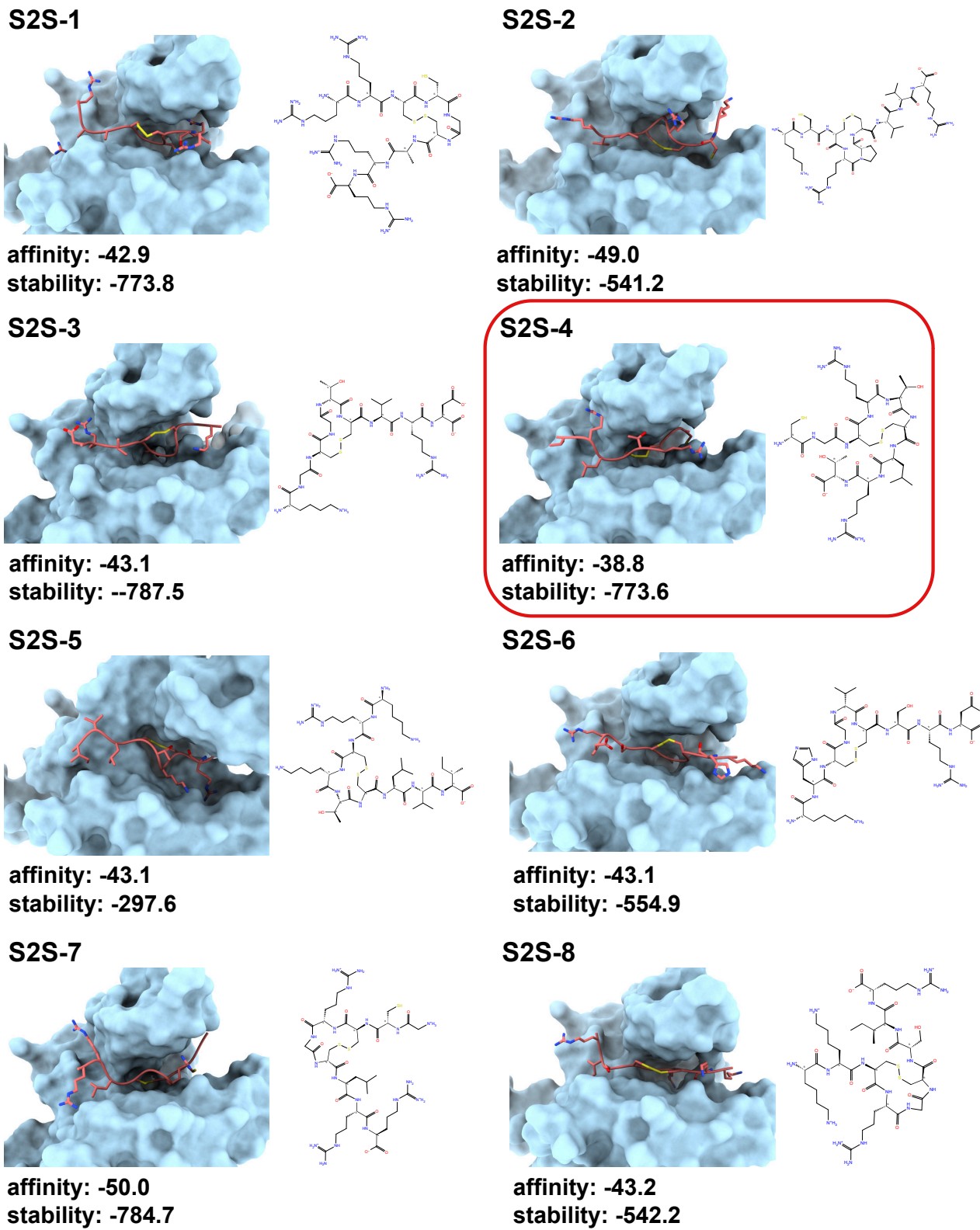

Figure 11. Side-to-side cyclic peptides designed for target SET8.

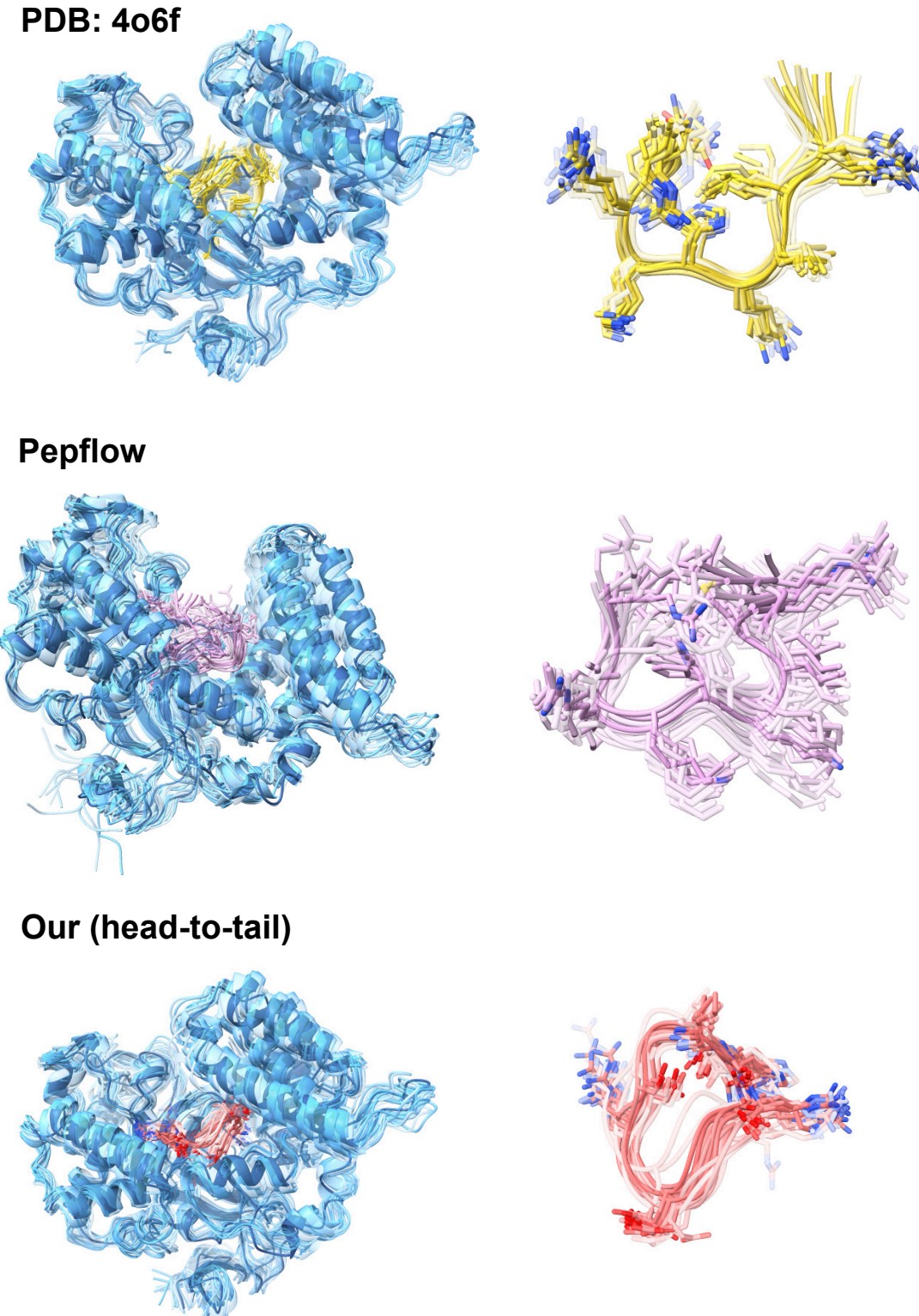

Figure 12. Structure ensembles of SMYD2 from multiple perspectives.

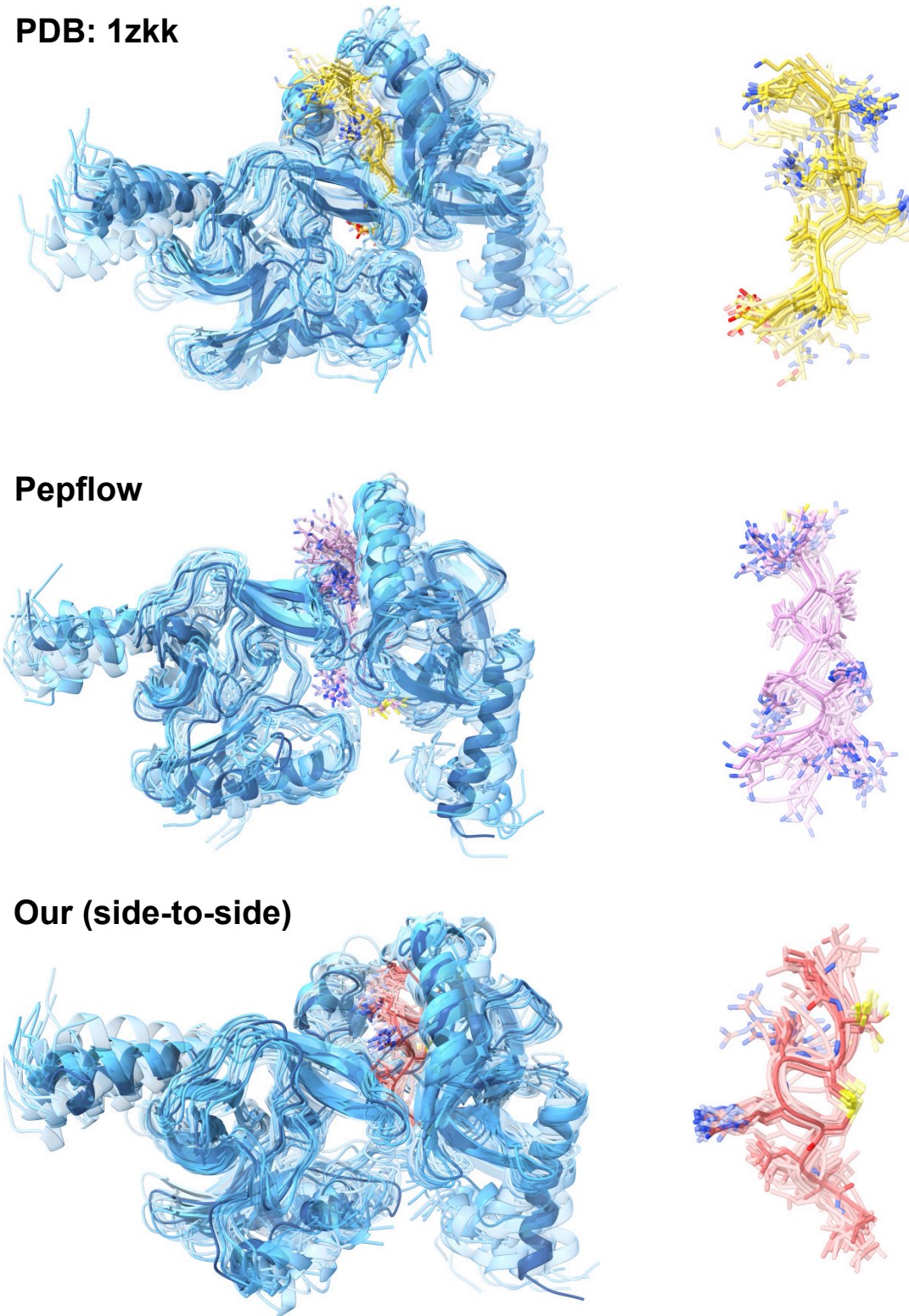

*Figure 13.* Structure ensembles of SET8 from multiple perspectives.

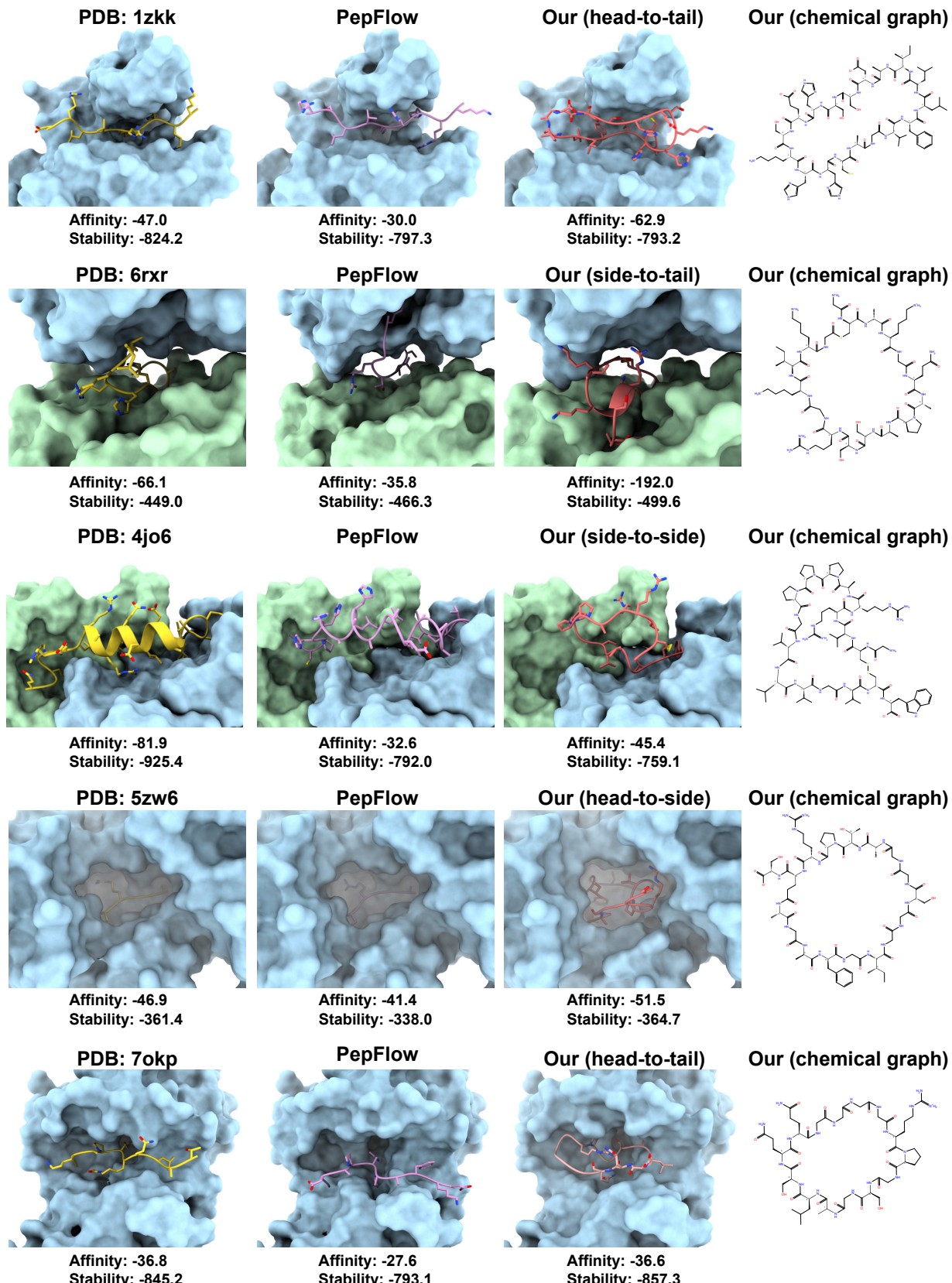

*Figure 14.* Examples of reference peptides, linear peptides designed by PepFlow, and cyclic peptides designed by our method.

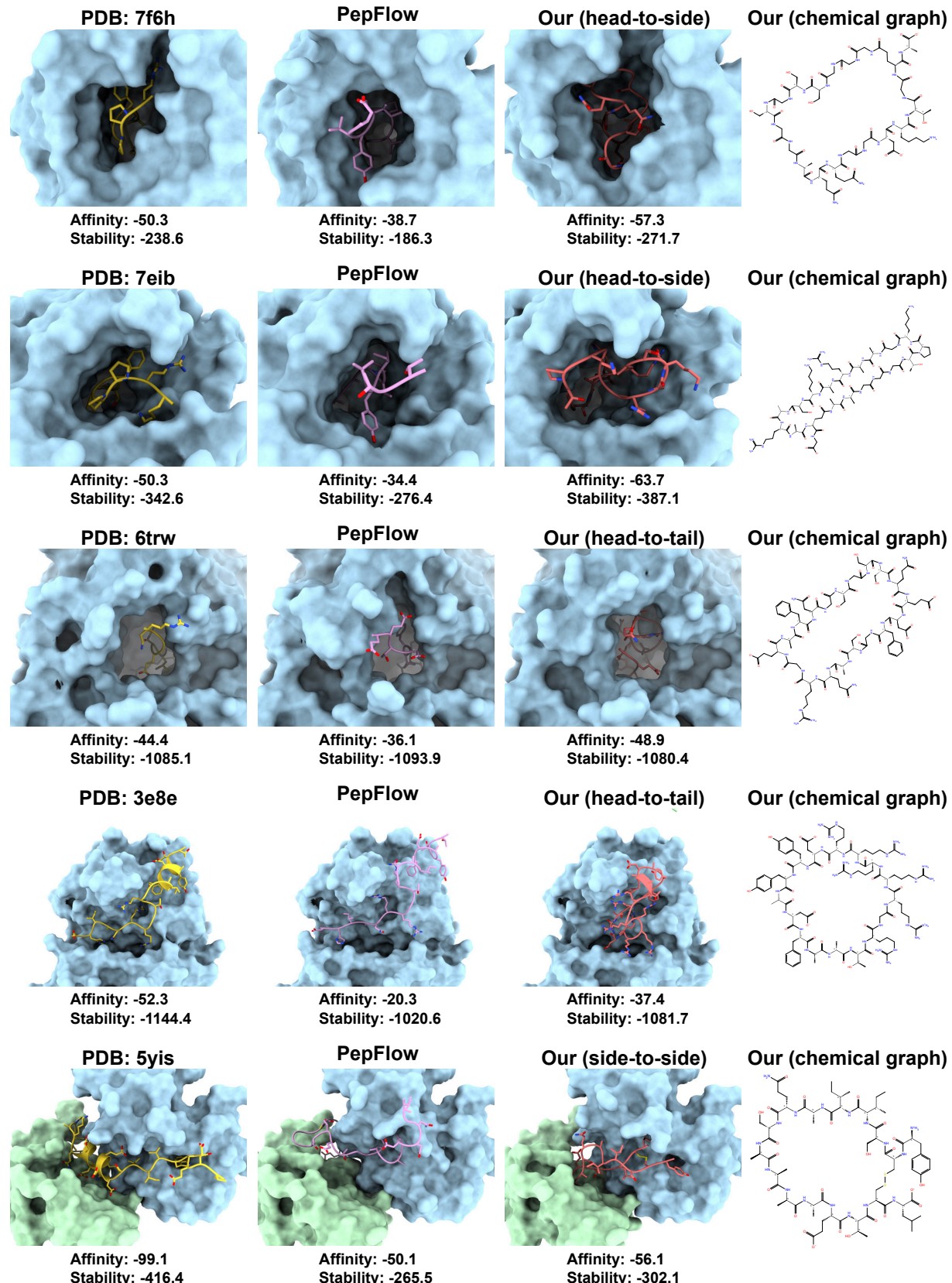

*Figure 15.* Examples of reference peptides, linear peptides designed by PepFlow, and cyclic peptides designed by our method.

