# OpenReview forum: "Designing Cyclic Peptides via Harmonic SDE with Atom-Bond Modeling"
_ICML.cc/2025/Conference — ICML 2025 poster_

### Official Review · Reviewer_NXAJ · 2025-03-01

**Overall Recommendation:** 3

**Summary:**

The paper introduces CPSDE, a new model for designing cyclic peptides using harmonic SDE and explicit atom-bond modeling conditioned on a 3D structure of a protein target. CPSDE comprises two key components: a generative structure prediction model and a residue type predictor. Alternating between these two models, CPSDE iteratively updates sequences and structures. CPSDE can be trained on small molecule and linear peptides, removing the need for abundant cyclic peptide data. It handles cyclization and non-standard amino acids. Experimental results show reliable stability and affinity, and some molecular dynamics simulations further validate the model in real-world design scenarios.

**Claims And Evidence:**

yes

**Essential References Not Discussed:**

no

**Experimental Designs Or Analyses:**

yes

**Methods And Evaluation Criteria:**

yes

**Other Comments Or Suggestions:**

/

**Other Strengths And Weaknesses:**

Strengths:
* the problem of cyclic peptide generation conditioned on a target structure is an important problem with few work able to handle both cyclization and non canoncial amino acid. as such the problem tackled is new from an application point of view. the fact that the model can be trained on non cyclic peptide is interesting as well.
* experimental results show that CPSDE outperforms baseline methods of linear peptides in terms of stability, affinity, and diversity. the authors also performed some MD simulations

Weakness:
* the main weakness I see for this paper in a ML venue is that it is highly specialized to tackle the problem of cyclic peptide design; arguably this is an important application, but the authors seem to propose a rather complex model to handle the cyclic peptide generation task.
* moreover, I understand there is not many work existing for cyclic peptide generation but the experimental comparison is done against approaches for linear peptide generation. it is thus complicated to understand how the proposed ML model compares to alternatives. could the authors provide a baseline (be it a non ML approach) to compare the performance ? I am aware of two work for cyclic peptide generation that are very recent (as such the authors are not required to include them for comparison)) e.g. PepTune: De Novo Generation of Therapeutic Peptides with Multi-Objective-Guided Discrete Diffusion Tang et al. and Accurate de novo design of high-affinity protein binding macrocycles using deep learning Rettie et al. although the latter cannot model non canonical amino acids; it would be helpful to compare a benchmark on a setting which can be handled by other models than CPSDE.

**Questions For Authors:**

* How does the generation time compare to alternative peptide generative models ? given all-atom and bond modeling, the sampling time might be heavy.




---- Post rebuttal ----

thank you for your rebuttal; I increased my score; the additional comparison on linear peptide generation is useufl in general to understand how this model compares to other models be it on non cyclic peptides.

**Relation To Broader Scientific Literature:**

Current methods for peptide design, mainly focused on linear peptides as they cannot include non canonical amino acids or are not designed to handle the cyclicity constraint. CPSDE addresses these limitations with atom-bond modeling.

**Theoretical Claims:**

no theoretical claims

---

> ### Author Rebuttal · Authors · 2025-04-01
>
> **Q1: "The main weakness I see for this paper in an ML venue is that it is highly specialized to tackle the problem of cyclic peptide design; arguably this is an important application, but the authors seem to propose a rather complex model to handle the cyclic peptide generation task."**
>
> A1: Thank you for recognizing the significance of cyclic peptide design. Our method is complex due to two main challenges: limited data on cyclic peptides; common protein representation based on residue frames that inadequately handle unique geometrical constraints and occasional non-canonical amino acids.
>
> Our approach addresses these issues by employing all atom and bond modeling and two integrated modules: AtomSDE and ResRouter. Both are essential and non-redundant. To our knowledge, this is the first generative model for designing cyclic peptides. We hope this sparks further research and advancements in this important field.
>
> **Q2: Could the authors provide a baseline (be it a non ML approach) to compare the performance? I am aware of two works for cyclic peptide generation that are very recent (as such the authors are not required to include them for comparison)) e.g. [1,2] although the latter cannot model non canonical amino acids.**
>
> Thank you for highlighting this. Both [1] and [2] are awesome works, yet they significantly differ from our approach.
>
> - [1] is a ligand-based drug design (LBDD) method that models the sequence of cyclic peptides using discrete diffusion, optimized by multiple reward functions. It doesn't explicitly incorporate the 3D structure of target proteins, whereas our structure-based drug design (SBDD) method directly designs ligands based on 3D target structures.
>
> - [2] uses modified RoseTTAFold and RFdiffusion with cyclic relative positional encoding to generate macrocyclic backbones. We've already cited [2] in our paper. It only supports head-to-tail cyclization due to its residue-level encoding limitations, whereas our work accommodates all four types of cyclic peptides.
>
> Both works are very recent: [1] was released on November 18, 2024, and [2] on December 23, 2024. As neither has released their code, we are unable to directly compare methods at this time. We will cite these papers and discuss them further in future versions of our paper.
>
> References:
>
> [1] PepTune: De Novo Generation of Therapeutic Peptides with Multi-Objective-Guided Discrete Diffusion, Tang et al.
>
> [2] Accurate de novo design of high-affinity protein binding macrocycles using deep learning, Rettie et al.
>
> **Q3: "It would be helpful to compare a benchmark on a setting which can be handled by other models than CPSDE."**
>
> A3: Given that all baseline methods, including ours, can design linear peptides, we compare them under these conditions. Please see our responses to Reviewer Rxgy's Q2 & Q3 for more details. Nonetheless, we continue to emphasize that the principal focus of our work is on cyclic peptide design.
>
> **Q4: "How does the generation time compare to alternative peptide generative models ? given all-atom and bond modeling, the sampling time might be heavy."**
>
> A4: We benchmark the average time of generating one peptide for all co-design baselines and our methods on a single NVIDIA A100-SXM4-80GB GPU. See the results below.
>
>
> | Method | Peptide Type | Time (s) |
> |---|---|---|
> | ProteinGenerator | Linear | 31.80 |
> | PepFlow | Linear | 12.09 |
> | PepGLAD | Linear | 4.40 |
> | CpSDE | Cyclic | 16.88 |
>
> Given that computational drug design does not demand real-time model response, the inference time of our method is deemed acceptable.

---

### Official Review · Reviewer_H4HQ · 2025-03-15

**Overall Recommendation:** 5

**Summary:**

This work tackles the task of cyclic peptide design. Cyclic peptides can have unique advantages in terms of stability and affinity when producing binders compared to other types of peptides or ligands. While there is much work in small molecule as well as protein and peptide generation, there is no prior work on producing cyclic peptides, a gap in the literature, which this paper fills. To this end, the authors propose CpSDE, a method consisting of AtomSDE, a structure prediction model, and ResRouter, a residue type predictor. The two components are called in an alternating manner in a denoising diffusion framework to produce novel cyclic peptides. The approach leverages an explicit all-atom formulation and builds on the atom73 representation with a side chain superposition framework used in previous work. CpSDE also includes explicit bond modeling, and cyclization and target information is given as conditioning. The paper computationally validates the approach through energy-based metrics for stability and validity as well as diversity. Moreover, it runs molecular dynamics simulation for selected cases, showing stable conformations of the generated cyclic peptides, thereby supporting high binding affinity.

**Claims And Evidence:**

All claims made in the paper are appropriately supported through convincing experiments.

**Essential References Not Discussed:**

I was not able to identify any essential work that was not cited.

**Experimental Designs Or Analyses:**

All experimental designs and analyses seem sound and valid to me. I have no concerns.

**Methods And Evaluation Criteria:**

All methods and evaluation criteria are appropriate for the problem at hand.

**Other Comments Or Suggestions:**

It would be great if the authors would release their curated training dataset as well as models and code for the broader community.

Moreover, I have some minor wording comments:
- Line 124, "...a groundbreaking approach...": I believe it is not up to the authors to decide themselves whether their approach is groundbreaking or not. "Groundbreaking" is a very strong word. The community will decide this. I would suggest to change this wording.
- Line 430, Conclusions, "...CpSDE is a pioneering...": Same issue, please tone down the wording and let the community judge whether this is pioneering or not.

**Other Strengths And Weaknesses:**

**Strengths:**
- To the best of my knowledge, this is the first protein/peptide design paper that tackles cyclic peptide generation, thereby filling a gap in the literature. This means that the work can be considered impactful and significant.
- The chosen methodology relies on existing techniques (harmonic SDE diffusion, graph neural networks, atom73 representation, etc.), which themselves are not novel, but these components are put together in a novel, original and well-motivated way for the task at hand.
- The quantitative comparisons to existing works show that CpSDE performs on-par with previous works. While one may criticize the work for not achieving state-of-the-art performance across the board on all metrics, all existing works only generate linear peptides (and in the case for RFDiffusion only generate non-diverse simple helices). CpSDE opens up the possibility for cyclic peptides, in contrast to all existing works, which is very innovative.
- The additional validation based on molecular dynamics simulation that goes beyond simple energy and diversity metrics is very nice and convincing.
- The paper is very well written and clearly explained, with an excellent introductory section, motivating cyclic peptide design and introducing it in an appropriate manner to the machine learning audience.
- The quality of the figures and visualizations is excellent.
- As discussed above, the supplementary material is very comprehensive and leaves no questions open.

**Weaknesses:**
Frankly, this is a great paper, which I enjoyed reading and reviewing, and I was not able to identify any major flaws or weaknesses that would make me question the work.

Consequently, I applaud the authors to their great work and highly recommend the paper for acceptance.

**Questions For Authors:**

- Line 207: What exactly is the role of $\sigma_P^{-2}$, when calculating $\boldsymbol{H}$? The paper only says that this is a "receptor-dependent scalar value", but no intuitions are given. It would be great if the authors could explain this better.

**Relation To Broader Scientific Literature:**

The authors did an excellent job putting the paper in the context of the broader literature and motivating their approach. The paper has a long list of references and an additional discussion of related work in the appendix.

**Theoretical Claims:**

The paper does not have any complex theorems or proofs, so this question does not apply. The maths around the harmonic SDE and the diffusion framework seems correct.

---

> ### Author Rebuttal · Authors · 2025-04-01
>
> **Q1: "It would be great if the authors would release their curated training dataset as well as models and code for the broader community."**
>
> A1: We would like to open source our work to contribute to the community.
>
> **Q2: "I have some minor wording comments."**
>
> A2: Thanks for pointing this out. We will change these words and choose more objective words in the future version of our paper.
>
> **Q3: "What exactly is the role of $\sigma_P^{-2}$, when calculating $\mathbf{H}$?"**
>
> A3: $\mathbf{H}=\mathbf{L}+\sigma_P^{-2}\mathbf{I}$. $\mathbf{L}$ is the Laplacian matrix. An intuitive explanation is that $\mathbf{L}$ encourages the connected atoms in the graph to be initialized closer. $\sigma_P$ is the standard derivation of the atom coordinates of the protein pocket.  An intuitive explanation is that $\sigma_P^{-2}\mathbf{I}$ encourages the atoms to be initialized more scattered when the pocket itself is large. This reflects a useful prior knowledge, as a pocket typically accommodates a ligand that complements its shape.

---

### Official Review · Reviewer_NNQv · 2025-03-17

**Overall Recommendation:** 4

**Summary:**

This paper describes a generative method to design cyclic peptides given a protein target. The method uses two diffusion models utilized in a coupled fashion. One to generate the structure, the other to predict the sequence.

## Update After Rebuttal
I thank the authors for addressing my review. I have decided to stay with my rating of 4.

**Claims And Evidence:**

The paper shows comparison with existing methods for linear peptides, and also shows predictions for some example targets. Overall, I think the evidence is adequate, but one evaluation that I would be interested in seeing is a comparison with a known therapeutic cyclic peptide. How well do the ligands generated by CpSDE compare against those?

**Essential References Not Discussed:**

N/A

**Experimental Designs Or Analyses:**

The experiment designs were adequate.I did not review them in depth, but referred to them to clarify some points.

**Methods And Evaluation Criteria:**

Covered above.

**Other Comments Or Suggestions:**

Figure 1 felt unnecessary, to me.

Overall, Figure 2 is not very clear, especially the denoising/renoising/ResRouter coupling. Seems to show repetitive denoising without re-noising. In 3.3, it is stated that the ligands are noisy. But in the figure, it shows ResRouter using denoised ligands.

Some intuition on why the renoising is necessary would be helpful.

**Other Strengths And Weaknesses:**

I think the work is innovative and interesting. I would have liked to see more convincing evidence that the benefits have real-world applicability, including, as mentioned above, comparing the generated ligands to known cyclic peptides.

**Questions For Authors:**

* AtomSDE only includes the protein target. How are different pockets specified?
* Page 2, line 191: The sentence beginning “The inclusion…” could be explained a bit more.
* Page 3, line 154: Why are the 3D structures of the cyclization part unavailable?
* In 4.1, it is not clear to me how the training/test splits are generated using the sequence identity.
* Can CpSDE be adapted for use on linear peptides? This would allow for a head-to-head comparison against other methods.

**Relation To Broader Scientific Literature:**

The key contributions were well-situated within the existing literature in the Related Work section.

**Theoretical Claims:**

N/A.

---

> ### Author Rebuttal · Authors · 2025-04-01
>
> **Q1: "Comparing the generated ligands to known cyclic peptides."**
>
> A1: We conducted a comparison of our method with known cyclic peptides.
>
> Vasopressin, a natural cyclic peptide featuring intramolecular disulfide (S-S) bond cyclization, is utilized in the treatment of antidiuretic hormone deficiency, vasodilatory shock, gastrointestinal bleeding, ventricular tachycardia, and ventricular fibrillation [1]. We selected two vasopressin-protein complexes, PDB IDs: 1JK4 and 1YF4, to design cyclic peptides targeting bovine neurophysin II and trypsin, respectively.
>
> The LEDGF binding site of HIV integrase (HIV-IN) represents a promising target for novel inhibitor development. Prior research has leveraged solution cyclization to discover various head-to-tail cyclic peptides interacting with this site [2]. The PDB IDs are 3A\*\* in the following table.
>
> The results are shown as follows:
>
> |PDB|Ref.||Our||
> |---|-|-|-|-|
> ||Stab.|Affi.|Stab.|Affi.|
> |1jk4|-307.23|-43.17|-215.91|-96.67|
> |1yf4|-52.36|-37.60|-47.03|-46.74|
> |3ava|-276.09|-30.49|-246.03|-31.98|
> |3avb|-248.83|-32.31|-233.89|-31.86|
> |3avg|-314.92|-31.15|-276.46|-37.93|
> |3avh|-278.42|-32.41|-255.95|-31.74|
> |3avi|-252.17|-42.72|-222.94|-33.89|
> |3avj|-381.38|-41.21|-334.38|-30.35|
> |3avk|-308.66|-38.18|-281.07|-30.07|
> |3avl|-306.86|-30.78|-220.60|-104.93|
> |3avm|-252.95|-31.78|-227.98|-26.94|
> |3avn|-262.63|-29.71|-228.48|-32.44|
>
> The results demonstrate that our method successfully designs cyclic peptides with affinity and stability comparable to known cyclic peptides, albeit with slightly lower stability than the reference.
>
> [1] Vasopressin: physiology, assessment and osmosensation, Bichet et al. Journal of Internal Medicine, 2017.
>
> [2] Crystal Structures of Novel Allosteric Peptide Inhibitors of HIV Integrase Identify New Interactions at the LEDGF Binding Site, Peat et al. ChemBioChem, 2011.
>
> **Q2: "Figure 1 felt unnecessary, to me."**
>
> A2: Figure 1 illustrates the therapeutic advantages of cyclic peptides compared to linear peptides. We would like to move this to the appendix.
>
> **Q3: Explain Figure 2, especially the denoising/renoising/ResRouter coupling. "Some intuition on why the renoising is necessary would be helpful."**
>
> A3: The generative process is a reverse SDE, where each Euler step consists of two parts: drift (denoising) and diffusion (renoising). See Equation (2) in the paper where the Wiener process introduces the stochasticity.
>
> In Figure 2, AtomSDE adjusts the Atom73 coordinates through denoising and renoising, treating the denoised ligand as an intermediate state. ResRouter alters the chemical graph by predicting amino acid types based on the denoised ligand, which provides more useful signals than the noised ligand. Importantly, ResRouter changes only the atom and bond types in the chemical graphs and leaves the atom73 coordinates unchanged. Similar operations are utilized in [3].
>
> [3] An all-atom protein generative model, Chu et al. PNAS, 2024.
>
> **Q4: "AtomSDE only includes the protein target. How are different pockets specified?"**
>
> A4: AtomSDE includes both pockets and noisy ligands. Our method aims to design cyclic peptides based on a given binding site. We refer to the protein target as the pockets. This setup aligns with the baselines, such as PepFlow. Specifically, the pockets are defined as residues of the protein target within 10 Angstroms surrounding a known ligand. We will include more details and clarifications in future version of our paper.
>
> **Q5: Explain page 2, line 091.**
>
> A5: Our method models all atoms and bonds, treating linear and cyclic peptides equivalently since both are composed of the fundamental components—atoms and bonds. Assuming AtomSDE is a well-trained docking model, it facilitates the generative process by encouraging two atoms connected by a chemical bond to be positioned closer together appropriately. By predetermining the cyclization type, we specify the related atom and bond types accordingly. Incorporating bond modeling ensures cyclization occurs naturally in 3D space, as bound atoms are drawn closer during the generative process.
>
>
> **Q6: Page 3, line 154: Why are the 3D structures of the cyclization part unavailable?**
>
> A6: The 3D structures of the cyclization segment are unavailable because they belong to what we aim to design.
>
> **Q7: "In 4.1, it is not clear to me how the training/test splits are generated using the sequence identity."**
>
> A7: Samples are grouped by receptor sequence similarity of 0.3 to create separate training and validation sets. In other words, if two samples are more than 30% similar in sequence, they cannot be in the same set. When receptors have multiple chains, the sequences are concatenated together to determine similarity. This method is widely used in previous works.
>
> **Q8: "Can CpSDE be adapted for use on linear peptides? This would allow for a head-to-head comparison against other methods.**
>
> A8: Please refer to our responses to Reviewer Rxgy's questions 2 and 3.

---

### Official Review · Reviewer_Rxgy · 2025-03-20

**Overall Recommendation:** 3

**Summary:**

The paper proposes an approach for the design of cyclic peptides using score-based generative models and diffusion. It is termed harmonic SDE, mainly because of conditioning on chemical graph that gives rise to a slightly non-standard forward process. The approach has been evaluated in peptide design against few other approaches, but with a caveat that their outputs are linear rather than cyclic peptides. The main aspects in empirical evaluation deal with stability, affinity, and diversity.

## POST-REBUTTAL ##
Thank you for the hard work. I will increase my score but please do a meaningful revision of the final paper.

**Claims And Evidence:**

I find the idea of conditioning generative models using chemical graphs interesting, and also the problem of designing cyclic peptides highly relevant for therapeutics purposes. However, the empirical evaluation of both ideas is inadequate. The experiments essentially compare apples-to-oranges in assessing designs that are linear vs cyclic peptides. What would make sense is to have the approach first design only linear peptides and evaluate the extra boost that comes as a result of conditioning on chemical graphs relative to several different architectures/baselines. My understanding is that data in crystal form would also allow for evaluating structural properties and reporting against these metrics vs same approaches without such conditioning. The second aspect is merits of cyclic peptides design which is challenging to evaluate due to the lack of crystal structures of complexes.

**Essential References Not Discussed:**

Good coverage of related work

**Experimental Designs Or Analyses:**

see above

**Methods And Evaluation Criteria:**

There are two components in the approach: i) ATOMSDE that is a docking model trained using both small molecules and peptides as ligands. It is unclear what fraction of data is actually cyclic peptides and why this dataset would be relevant for generating from that class. ii) RESROUTER that predicts amino-acid from aggregated hidden stats representing backbone atoms. Still the question is if the data on cyclic peptides is sparse how the model can mine useful signal for completing cyclicazation.

Please see above for more details on evaluation.

**Other Comments Or Suggestions:**

see above

**Other Strengths And Weaknesses:**

see above

**Questions For Authors:**

see above

**Relation To Broader Scientific Literature:**

Good coverage of related work on generative models for peptides

**Theoretical Claims:**

Not applicable

---

> ### Author Rebuttal · Authors · 2025-04-01
>
> **Q1: "I find the idea of conditioning generative models using chemical graphs interesting, and also the problem of designing cyclic peptides highly relevant for therapeutics purposes. However, the empirical evaluation of both ideas is inadequate."**
>
> A1: Our work focuses on designing cyclic peptides instead of linear peptides, although our method is indeed capable of designing the latter as well. Our method is tailored specifically for cyclic peptide design. Designing linear peptides often involves modeling translation, rotation of residue frames, and side-chain torsion angles—techniques that essentially encompass all-atom coordinate modeling, as seen in methods like PepFlow. However, these techniques fall short for cyclic peptide design due to unique geometrical constraints and non-canonical amino acids involved. This challenge motivated us to introduce two modules: AtomSDE and ResRouter. These modules enable explicit modeling of all atoms and bonds, addressing the limitations of previous methods in cyclic peptide design. In the generative process, AtomSDE adjusts atom coordinates based on chemical graphs, while ResRouter refines these graphs by predicting amino acid types. We hope this explanation clarifies the motivation and contributions of our work.
>
> Besides, we have ablated the effect of Harmonic AtomSDE and ResRouter in designing cyclic peptides. Please refer to Appendix H.3.
>
> **Q2: Evaluation of extra boost in linear peptide design.**
>
> A2: While designing linear peptides is not our primary focus, we appreciate your suggestion to compare our method with existing baselines in this task.
>
> |Method|Co-Design|Peptide Type|Stability||Affinity||Diversity|
> |---|-|-|-|-|-|-|-|
> ||||Avg.|Med.|Avg.|Med.||
> |Reference||Linear|-672.53|-634.71|-85.03|-78.70||
> |RFDiffusion|N|Linear|-633.51|-607.82|-70.30|-61.35|0.55|
> |ProteinGenerator|Y|Linear|-576.39|-554.70|-46.98|-40.39|0.58|
> |PepFlow|Y|Linear|-576.16|-498.31|-47.88|-42.40|0.70|
> |PepGLAD|Y|Linear|-359.44|-310.33|-45.06|-38.56|0.79|
> |CpSDE|Y|Linear|-567.34|-510.58|-55.48|-49.89|0.77|
>
> Our method excels in designing linear peptides with superior affinity and comparable stability among all co-design methods, while also maintaining considerable diversity. Although our method slightly lags behind RFDiffusion, it's worth noting that RFDiffusion often generates helices and relies (also observed for ProteinGenerator) on a two-stage design pipeline (first designing the backbone, then the sequence).
>
> **Q3: "Data in crystal form would also allow for evaluating structural properties and reporting against these metrics." "The second aspect is merits of cyclic peptides design which is challenging to evaluate due to the lack of crystal structures of complexes."**
>
> A3:  Evaluating energy is more critical than assessing RMSD against crystal structures of known binders, as there can be numerous design alternatives and a low RMSD does not always indicate a superior design model [1]. Therefore, we focus primarily on evaluating stability and affinity from an energy perspective. To further validate our findings, we conduct Molecular Dynamics simulations.
>
> We have also designed linear peptides as in Q1&A1 and computed the RMSD against known linear peptide binders, shown as follows:
>
> |Method|Co-Design|Peptide Type|RMSD(Avg.)|RMSD(Med.)|
> |---|-|-|-|-|
> |RFDiffusion|N|Linear|2.85|2.27|
> |ProteinGenerator|Y|Linear|3.54|3.12|
> |PepFlow|Y|Linear|1.60|0.98|
> |PepGLAD|Y|Linear|2.17|1.19|
> |CpSDE|Y|Linear|1.43|1.17|
>
> Our method achieves the lowest average RMSD among all methods, along with a comparable median RMSD.
>
> [1] Antigen-specific antibody design via direct energy-based preference optimization, Zhou et al. NeurIPS 2024.
>
> **Q4: "What fraction of data is actually cyclic peptides and why this dataset would be relevant for generating from that class." "If the data on cyclic peptides is sparse how the model can mine useful signal for completing cyclicazation."**
>
> A4: Our method successfully designs cyclic peptides despite the training set containing less than 6% cyclic peptide data. This achievement stems not from relying on this small percentage, but from our approach of explicitly modeling all atoms and bonds, surpassing previous methods in depth. Through this fine-grained model, linear and cyclic peptides are treated equivalently as both are composed of the same fundamental components: atoms and bonds. The cyclization type is predetermined, allowing us to specify the related atom types and bond types accordingly. By incorporating bond modeling, we ensure cyclization occurs in 3D space since bound atoms are naturally drawn closer during the generative process. The ResRouter model equally sidesteps dependency on scarce cyclic peptide examples due to its ability to deduce amino acid types based on the contextual atom arrangement within the peptide and its receptor. Thus, it is this distinctive and comprehensive modeling technique that enables overcoming data limitations in cyclic peptide design.

---

> > ### Comment · Reviewer_Rxgy · 2025-04-02
> >
> > I would like to thank the authors for additional experiments. I have read the rebuttal and revisited the paper once more. My estimate is that the paper requires a revision that goes beyond what can be done during the rebuttal period and as a result I have decided to keep my original rating.
> >
> > **Re: cyclic peptides and non-canonical amino-acids**\
> > Hit-to-lead nomination with cyclic peptides is typically done exclusively with natural amino-acids, and roughly once one reaches single digit nM or even sub-nM range only then non-naturals are taken into consideration to tackle potential enzyme cleavage sites, improve stability, slow down degradation by immune system, etc. Generative models are typically unable to deliver that potency range and, thus, restricting to cyclic peptides with natural amino-acids would be reasonable.
> >
> > **Re: linear vs cyclic peptides in the original experiments**\
> > Let me now get back to the main concern, which is evaluation and the fact that the original submission compares apples-to-oranges, as cyclic peptides are more rigid than linear ones and, thus, have less conformational freedom leading to stronger and more specific interactions with target molecules. Hence, selected metrics are expected to show better results for methods that design cyclic peptides (i.e., proposed approach), relative to outputs of baselines that design linear peptides.
> >
> > **Re: metrics**\
> > The two key metrics in the original submission are stability and affinity, which deal with total energy of reference ligands and interface binding energies from Rosetta (Chaudhury et al., 2010). In benchmarking relative to Kd from SPR or HTRF assays, the latter does not fare well and on average Pearson correlation is around 10% (up to 20% at best). Hence, a fair question would then be why this would be a good metric for assessing the quality of generative models.
> >
> > The main advantage in comparison to RFDiffusion (on linear peptides) appears to be diversity. However, it is encouraging to see good results on RMSD for linear peptides. Additional aspects that would be interesting are binding site ratio (BSR), fraction of hydrophobic and/or charged residues (relevant for specificity), l-RMSD and i-RMSD, differences in lengths and angles (e.g., see Lin et al., ICML 2024).
> >
> > **Re: data splits**\
> > Given that there is 6% or so of structures with cyclic peptides, it would be interesting to see structural metrics relative to some fraction of these. It would also be relevant to report structural and sequence similarities of binding pockets relative to training sample.

---

> > > ### Author Response · Authors · 2025-04-07
> > >
> > > Thanks for your comment.
> > >
> > > **Q1: cyclic peptides and non-canonical amino-acids**
> > >
> > > A1: Some NCAAs are indeed introduced post-hoc. Notably, our method **does not contradict with** your claim that "restricting to cyclic peptides with natural amino-acids would be reasonable", highlighted by:
> > > 1. NCAAs are included only in the cyclization segment as constraints, such as in the side-to-side cyclization in 1RGR\_B (Figure 6).
> > > 2. ResRouter only predicts one of the 20 canonical amino acids for free residues (see Equation 5 and Section 3.1). Four demonstrative cyclic peptide types in our experiments do not include NCAAs: head-to-tail (N-term and C-term C-N bond), head-to-side (first residue's N to side-chain C), side-to-tail (CYS side-chain S and backbone C form C-S bond), and side-to-side (two S in CYS side chain form S-S bond). Figure 9 shows these examples of cyclic peptides designed by our method with only canonical amino acids. The cases used for MD simulations also only involve natural amino acids.
> > >
> > > **Q2: linear vs cyclic peptides in the original experiments**
> > >
> > > A2: It is true that cyclic peptides are usually more rigid than linear ones and have less conformational freedom leading to stronger and more specific interactions with target molecules. These properties underscore the natural advantages of valid and reasonable cyclic peptides. We firmly believe this isn't an aspect of unfair comparison, as randomly designed cyclic peptides do not inherently have these characteristics. Please see the ablation studies in the appendix. Additionally, we compared our designed cyclic peptides with known cyclic peptide binders, and the experiments demonstrated that the designed peptides displayed reasonable stability and affinity. For more information, please see our response to Reviewer NNQv's Q1.
> > >
> > > **Q3: metrics**
> > >
> > > A3: Rosetta is a widely used and effective tool for in-silico evaluations, with many studies using its energy scores as a key metric, e.g., [1,2]. We also ran **MD simulations, which are more reliable but expensive**, to validate our designed cyclic peptides.
> > >
> > > We have done more comprehensive evaluation on linear peptides as you required, as follows:
> > > |Method|Hydrophobic Ratio|Charged Ratio|DockQ|iRMSD|LRMSD|BSR|
> > > |---|---|---|---|---|---|---|
> > > |Reference|0.48|0.28|||||
> > > |ProteinGenerator|0.53|0.32|0.12|5.56|23.97|0.20|
> > > |PepFlow|0.60|**0.17**|**0.44**|2.49|**9.42**|0.56|
> > > |PepGLAD|0.53|0.25|0.30|2.68|11.99|0.39|
> > > |RFDiffusion|0.59|0.27|0.18|5.37|20.10|0.33|
> > > |CpSDE (linear)|**0.45**|0.24|0.32|**2.36**|9.91|**0.60**|
> > >
> > > We use DockQ package (https://github.com/bjornwallner/DockQ) to compute DockQ, iRMSD, and LRMSD. We follow the definition of binding site ratio (BSR) in [1].
> > > A lower hydrophobic/charged ratio indicates a lower risk of non-specific binding [3]. Notably, the fraction of hydrophobic and/or charged residues of our designed peptides resemble that of reference. Our method also shows superiority in structural properties. We also report Jensen–Shannon divergence (JSD) between designed linear peptides and reference linear peptides in https://anonymous.4open.science/r/cpsde_rebuttal-3578/bond_length_and_angle.md.
> > >
> > > **Q4: data split**
> > >
> > > A4: Our model understands bonds and atoms but does not differentiate between "linear" and "cyclic" peptides. As a result, from the model's perspective, these peptides are essentially the same. We evaluated the TM score of the designed peptides against those from the training set across four typical types of cyclic peptides. The results confirm that the generated peptides do not closely resemble any within the training set.
> > >
> > > |Cyclic type|Number of  peptides in training set|TM score|||
> > > |---|---|:---:|:---:|:---:|
> > > |||Max|Average|Median|
> > > |Head-to-tail|61|0.334|0.173|0.168|
> > > |Head-to-side|31|0.350|0.176|0.175|
> > > |Side-to-tail|69|0.348|0.183|0.187|
> > > |Side-to-side|1001|0.429|0.198|0.196|
> > >
> > > For similarity of receptors, please refer to Section 4.1 and our response to Reviewer NNQv's Q7.
> > >
> > > **References:**
> > >
> > > [1] Full-Atom Peptide Design based on Multi-modal Flow Matching, Li et al. ICML 2024.
> > >
> > > [2] Antigen-specific antibody design via direct energy-based preference optimization, Zhou et al. NeurIPS 2024.
> > >
> > > [3] Optimization of therapeutic antibodies for reduced self-association and non-specific binding via interpretable machine learning, Makowski et al. Nature Biomedical Engineering 2023.

---

### Decision · Program_Chairs · 2025-05-01

**Decision:**

Accept (poster)

**Comment:**

Summary:

The paper introduces CPSDE, a new model for designing cyclic peptides using harmonic SDE and explicit atom-bond modeling conditioned on a 3D structure of a protein target. CPSDE comprises two key components: a generative structure prediction model and a residue type predictor. Alternating between these two models, CPSDE iteratively updates sequences and structures. CPSDE can be trained on small molecule and linear peptides, removing the need for abundant cyclic peptide data. It handles cyclization and non-standard amino acids. Experimental results show reliable stability and affinity, and some molecular dynamics simulations further validate the model in real-world design scenarios.

Strengths:

- The work is innovative and interesting.
- Idea of conditioning generative models using chemical graphs is interesting.
- The problem of designing cyclic peptides highly relevant for therapeutics purposes.
- First protein/peptide design paper that tackles cyclic peptide generation.
- The chosen methodology relies on existing techniques but these components are put together in a novel, original and well-motivated way for the task at hand.
- The quantitative comparisons to existing works show that CpSDE performs on-par with previous works. While one may criticize the work for not achieving state-of-the-art performance across the board on all metrics, all existing works only generate linear peptides (and in the case for RFDiffusion only generate non-diverse simple helices). CpSDE opens up the possibility for cyclic peptides, in contrast to all existing works, which is very innovative.
- The additional validation based on molecular dynamics simulation that goes beyond simple energy and diversity metrics is very nice and convincing.
- The paper is very well written and clearly explained, with an excellent introductory section, motivating cyclic peptide design and introducing it in an appropriate manner to the machine learning audience.
- The quality of the figures and visualizations is excellent.
- The supplementary material is very comprehensive and leaves no questions open.
- The problem of cyclic peptide generation conditioned on a target structure is an important problem with few work able to handle both cyclization and non canoncial amino acid. as such the problem tackled is new from an application point of view.
- The fact that the model can be trained on non cyclic peptide is interesting as well.
- Experimental results show that CPSDE outperforms baseline methods of linear peptides in terms of stability, affinity, and diversity. the authors also performed some MD simulations

Weaknesses:

- The experiments essentially compare apples-to-oranges in assessing designs that are linear vs cyclic peptides.
- Merits of cyclic peptides design is challenging to evaluate due to the lack of crystal structures of complexes.
- A meaningful revision of the final paper is recommended by reviewer Rxgy.
- Would have liked to see more convincing evidence that the benefits have real-world applicability.
- Would have been good to compare the generated ligands to known cyclic peptides (the authors provided additional evidence in rebuttal).
- The authors seem to propose a rather complex model to handle the cyclic peptide generation task.
- The experimental comparison is done against approaches for linear peptide generation.

Recommendation:

All the reviewers vote for acceptance. I, therefore, recommend to accept the paper and encourage the authors to use the feeback provided to improve the paper for its camera ready version.